# PAL: PROBING AUDIO ENCODERS VIA LLMS - AUDIO INFORMATION TRANSFER INTO LLMS

## ABSTRACT

Integration of audio perception into large language models (LLMs) is an emerging research area for enabling machine listening applications, yet efficient transfer of rich audio semantics from audio encoders to LLMs remains underexplored. The most widely used integration paradigm projects the audio encoder output tokens into the LLM input space (e.g., via an MLP or a Q-Former), then *prepends or inserts* them to the text tokens. We refer to this generic scheme as *Prepend to the LLM's input token space (PLITS)* integration. We propose an efficient alternative, Lightweight Audio LLM Integration (**LAL**). LAL introduces audio representations solely via the attention mechanism within different layers of the LLM, bypassing its feedforward module. LAL encodes rich audio semantics at an appropriate level of abstraction for integration into different blocks of LLMs. Our design significantly reduces computational overhead compared to existing integration approaches. Observing that Whisper style speech encoders benefit from PLITS integration, we propose an audio encoder aware approach for efficiently Probing Audio encoders via LLM (**PAL**), which in its multi encoder form employs PLITS for Whisper speech encoder and LAL for general audio encoders, and in its unified encoder form uses a single audio encoder but applies PLITS only to a compact set of speech summary tokens while integrating the full audio token sequence via LAL to preserve speech decoding capacity with low computational cost. Under an identical training curriculum, **LAL** consistently maintains performance or outperforms existing integration approaches across multiple base LLMs and tasks. For general audio tasks, LAL achieves improvements of up to 30% over a strong PLITS baseline, while reducing memory usage by about 60% and increasing throughput by about 190%. Furthermore, for general audio-music-speech LLM, **PAL**, performs on par with a fully PLITS integration-based system but with substantially improved computational and memory efficiency.

## 1 INTRODUCTION

Large Language Models (LLMs) (Brown et al., 2020; Grattafiori et al., 2024; Jiang et al., 2024; Liu et al., 2024a) have emerged as the foundational technology for natural language interaction with machines, demonstrating remarkable conversational fluency. Despite this success, their perceptual capabilities remain limited primarily to text, restricting their ability to understand the physical world. This limitation has inspired significant research into multi-modal LLMs (MLLMs), which expand traditional LLMs by integrating additional sensory modalities such as vision (Vision LLMs) (Liu et al., 2023; Templeton et al., 2024; Wang et al., 2024), audio (Large Audio Language Models (LALMs) or simply audio-LLMs) (Deshmukh et al., 2023; Gong et al., 2024; Tang et al., 2024; Ghosh et al., 2024; 2025a), and other inputs (Brohan et al., 2023; Thawkar et al., 2023) to foster more natural, intuitive, and effective human-machine interfaces.

An audio LLM typically comprises three components: (i) a large language model (LLM), (ii) one or more audio encoders, and (iii) a mechanism that integrates encoder outputs into the LLM. In this work, we investigate two such designs: a multi-encoder architecture that combines complementary encoders for general audio understanding (eg: Alex et al. (2025); Elizalde et al. (2023); Wu et al. (2023)) and speech understanding ( Radford et al. (2023)), and a unified architecture that has combined general audio speech understanding (AF-Whisper encoder in Goel et al. (2025)).

When it comes to the integration of audio encoders with the LLM, two architectural paradigms dominate today. The first transforms the outputs of an audio encoder or encoders into the LLM input space (e.g., via an MLP, a QFormer (Li et al., 2023), etc.), then *prepend or insert* these audio tokens to the text tokens and propagates the entire sequence through all LLM layers as if decoding jointly over audio and text. Please note that the common theme in this family is how audio tokens are passed to the LLM: they are *prepended* to the text tokens. We refer to this generic scheme **Prepend to the LLM's input token space (PLITS)** integration, a term we have introduced to group many state of the art methods in this family of audio LLMs such as Wu et al. (2025b); Xu et al. (2025); Chu et al. (2024); Goel et al. (2025); Chu et al. (2023); Ghosh et al. (2024); Tang et al. (2024); Gong et al. (2024); Deshmukh et al. (2023). The second paradigm, **Flamingo style** architectures (Alayrac et al., 2022; Kong et al., 2024), instead insert cross attention and feedforward (FFN) blocks *between* successive LLM layers; at each insertion, text tokens attend to a set of latent audio tokens, pass through the block FFN, and only then proceed to the next LLM layer. While this design improves attention efficiency relative to PLITS concatenation, the interleaved cross attention plus FFN stacks increase sequential depth and per layer compute, which can slow the forward pass.

In contrast, we introduce **LAL**, a lightweight integration that injects audio tokens into the LLM's attention blocks *as keys and values only* (without forming audio queries) and *bypasses the LLM FFNs for audio tokens*. This reduces the attention complexity from $\mathcal{O}\big((N_a+N_t)^2\big)$ to $\mathcal{O}\big((N_a+N_t)N_t\big)$, where $N_a$ and $N_t$ denote the numbers of audio and text tokens, respectively. Since typically $N_a \gg N_t$, this yields substantial efficiency gains. By avoiding both quadratic attention over audio tokens and their passage through LLM FFNs, LAL substantially reduces memory usage and computation. Unlike parameter-efficient methods such as LoRA, this is a core architectural modification, so the efficiency benefits are realized not only during training but also at inference time.

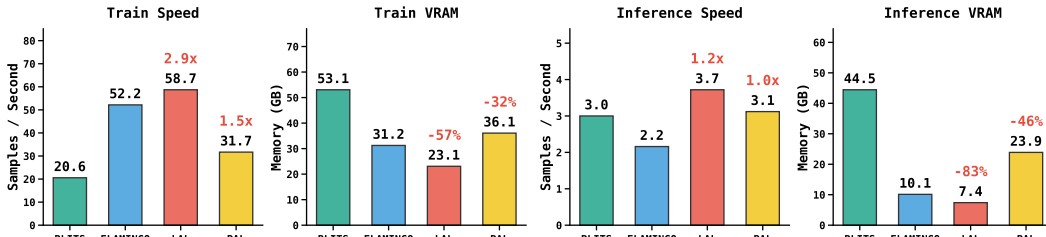

Figure 1: Comparison of compute efficiency between **PLITS, state of the art audio-LLM integration** (our baseline), **Flamingo**, **LAL**(ours) and **PAL**(Ours). All use Llama3.2-1B LLM and Multi audio encoder setup(ref to Section 3.3.1). Training was performed with batch size 8 on an NVIDIA A100 using bfloat16, and inference with batch size 12 on an NVIDIA A100 using float16. All benchmarks were executed sequentially on the same node to eliminate load-related discrepancies.

LAL provides a compute and memory-efficient mechanism by constraining how audio tokens interact with the LLM. However, some modalities, especially speech, which closely mirrors text, may benefit from the richer token-level decoding of PLITS-style integration. Motivated by this observation, we introduce two hybrid variants that combine LAL and PLITS, one in the multi audio encoder setting and one in the unified audio encoder setting. We refer to this family as the PAL framework for building general purpose audio, music, and speech LLMs, enabling fusion that balances efficiency and performance. This design achieves strong results while substantially reducing computational and memory requirements compared to using PLITS style integration alone.

To validate these architectural choices, we conduct a systematic empirical study under a standardized training curriculum and dataset setup, ensuring fair comparisons across models. Our experiments explore the trade-off between performance and efficiency, highlighting how different integration techniques facilitate effective information transfer from audio encoders to LLMs with minimal parameter overhead. This analysis provides actionable insights into the design of scalable and efficient audio LLMs that leverage diverse pretrained audio encoders.

**Our main contributions are as follows:** **(1)** We introduce **LAL**, a lightweight integration strategy for audio-LLMs that incorporates audio tokens solely as keys and values in the LLM's attention sub-modules and skips FFNs, thereby reducing computation and memory cost while retaining

performance comparable to PLITS integration, **(2)** Motivated by the observation that speech understanding benefits from PLITS integration, we propose **PAL**, a hybrid integrated LLM. In the multi encoder setup, PAL is encoder aware and selectively applies LAL or PLITS based on the audio encoder, while in the unified encoder setup it applies PLITS to a summarized subset of tokens and LAL to all tokens, enabling general purpose audio, speech, and music LLMs that balance efficiency and performance, and **(3)** We conduct **fair and rigorous architectural comparisons** under a standardized training curriculum and dataset setup, providing actionable insights into the efficiency–performance trade-offs of audio-LLM design.

## 2 LITERATURE REVIEW

**Audio LLM architectures:** When integrating audio encoders with an LLM, two paradigms dominate. In PLITS, encoder features are mapped to the LLM token space with a small projector such as an MLP or a Q Former, the resulting audio tokens are typically prepended to the text tokens, and the joint sequence is processed by all LLM layers (Wu et al., 2025b; Xu et al., 2025; Chu et al., 2024; Goel et al., 2025; Chu et al., 2023; Ghosh et al., 2024; Tang et al., 2024; Gong et al., 2024; Deshmukh et al., 2023). In contrast, the Flamingo style architecture inserts cross attention and feed forward adapters between successive LLM layers so that text tokens attend to latent audio tokens at selected depths (Alayrac et al., 2022; Kong et al., 2024). This makes audio to text interaction explicit and gated, but adds sequential depth, per layer compute, and parameters.

**Audio-LLM Datasets:** Beyond architecture, recent works have focused on high-quality instruction tuning datasets, both open-source and proprietary (Goel et al., 2025; Ghosh et al., 2024; Chu et al., 2024; Xu et al., 2025) and build audio reasoning benchmarks (Sakshi et al., 2024; Deshmukh et al., 2025a;b). Training PLITS or Flamingo-style models on these resources improves instruction following and audio reasoning, with most gains driven by the data rather than the integration scheme.

## 3 METHODOLOGY

This section outlines our approach to integrating audio with language models. We begin by formalizing **PLITS**, the SOTA audio-LLM integration, as our reference baseline. We then introduce **LAL**, a lightweight alternative that injects audio through attention only, and we analyze its compute and memory profile. Finally, we connect these findings to **PAL**, an encoder aware hybrid that selects between PLITS and LAL on a per encoder basis in order to support speech understanding without sacrificing efficiency on general audio.

### 3.1 BASELINE AUDIO LLM: PREPEND TO THE LLM'S INPUT TOKEN SPACE (PLITS)

To provide a fair comparison point for our integration methods, we construct a baseline audio LLM that follows the widely adopted SOTA integration strategy, which we refer to as *Prepend to the LLM's input token space (PLITS)*. In this design, the audio encoder outputs are first mapped into the LLM input embedding space using a Q-Former–style connector. The resulting audio tokens are then *prepended* to the text tokens, and the concatenated sequence is passed through all LLM layers so that decoding proceeds jointly over audio and text (see Fig. 2(A)).

The central characteristic of this PLITS-style integration is that **the audio tokens are *prepended* to the text tokens**. This integration strategy is used by most audio LLMs, including several state of the art systems Wu et al. (2025b); Xu et al. (2025); Chu et al. (2024); Goel et al. (2025); Chu et al. (2023); Ghosh et al. (2024); Tang et al. (2024); Gong et al. (2024); Deshmukh et al. (2023).

### 3.2 LAL: LIGHTWEIGHT AUDIO-LLM INTEGRATION

Recent work in mechanistic interpretability suggests that LLMs encode semantics as features that can be selectively activated within hidden states (Elhage et al., 2022; Bricken et al., 2023; Templeton et al., 2024). Building on this view, we hypothesize that effective audio LLM integration requires audio tokens to trigger the activation of sound related conceptual features inside the textual token embeddings. In other words, distinct auditory inputs should induce the corresponding linguistic concepts to become active in the text representation; for example, when the input contains a *dog bark*,

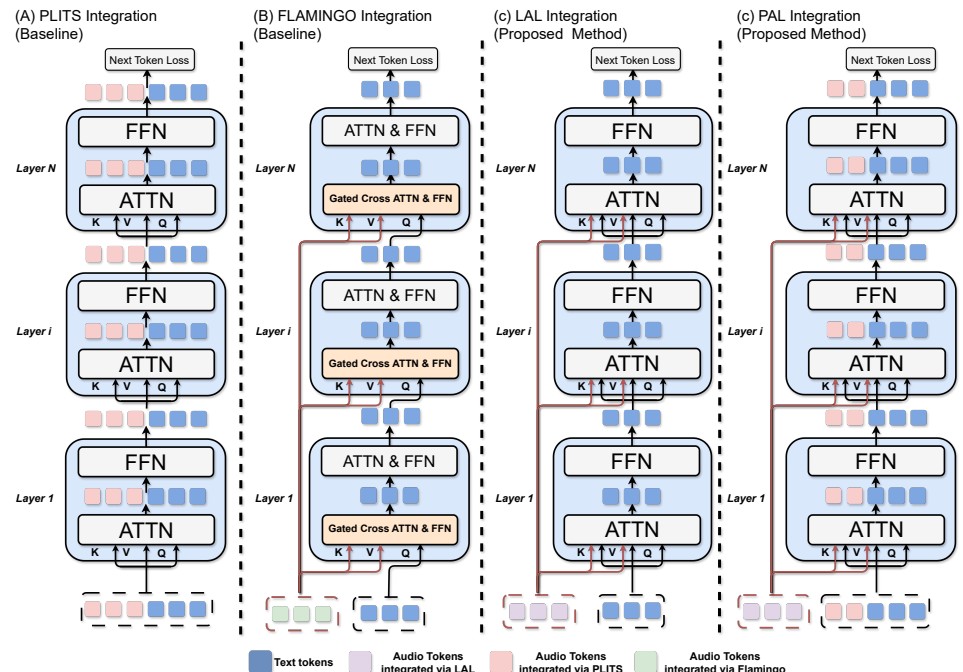

Figure 2: Illustration of integration techniques: (A) SOTA integration **PLITS** (prepend to the LLM's input token space), which prepends audio tokens to text tokens and propagates the full sequence through all LLM layers (our baseline);(B) **Flamingo** integration, where text tokens first attend to audio tokens through a separate cross attention plus FFN module, and the resulting signal is added to the text residual stream before the next LLM layer. (C) our proposed lightweight integration **LAL**, which introduces audio representations only through the attention mechanism (see Equations 2, 3, and 4) while bypassing the feedforward modules; (C) the hybrid **PAL**, an encoder aware/hybrid integration that combines **LAL** and **PLITS** integrations.

the features associated with the concept *dog* should light up so the model can ground the auditory signal in language and answer queries such as *Which animal sound is present?*. This hypothesis guides our architectural design: we seek the simplest pathway that reliably transmits audio cues into the text features that carry concepts.

A standard LLM layer consists of an attention submodule followed by a feed-forward network (FFN) submodule. Since attention mediates all inter-token interactions, it is the necessary pathway for audio to influence text, and we posit that it is also sufficient for text tokens to gather information from audio. Guided by this principle, we introduce **LAL** (Lightweight Audio LLM integration).

As in our baseline, a shared Q-Former produces a sequence of audio tokens and at each layer a small MLP projects these tokens into that layer's input space. Audio information is then injected into the attention block only through Keys and Values while Queries remain text only, so audio modulates the attention context of text tokens without passing through the feed-forward network.

Formally, let $H_l^t \in \mathbb{R}^{N_t \times d}$ denote the text hidden states at layer $l$ and $A \in \mathbb{R}^{N_a \times d_a}$ the Q-Former audio features. A per-layer projector $P_l : \mathbb{R}^{d_a} \to \mathbb{R}^d$ maps audio to the layer space,

$$\hat{A}_l = P_l(A) \in \mathbb{R}^{N_a \times d} \tag{1}$$

and we concatenate text and audio along the token axis

$$S_l = \left[ H_l^t ; \hat{A}_l \right] \in \mathbb{R}^{(N_t + N_a) \times d}. \tag{2}$$

Queries are formed from *text only* (see Figure 2(B)), while Keys and Values are computed from the concatenated sequence:

$$Q_l^t = H_l^t W_{Q,l}, \qquad K_l = S_l W_{K,l}, \qquad V_l = S_l W_{V,l}. \tag{3}$$

The resulting LAL update for text tokens is

$$\tilde{H}_l^t = \mathrm{softmax}\left(\frac{Q_l^t K_l^\top}{\sqrt{d_k}}\right) V_l. \tag{4}$$

after which $\tilde{H}_l^t$ proceeds through the FFN with the usual residual connections. In this way, audio cues shape the attention context seen by text tokens, aligning audio-evoked features with their linguistic counterparts and enabling effective cross–modal information transfer.

**Information Injection Dynamics. LAL is neither Lite PLITS nor Lite Flamingo.** Beyond computational efficiency, LAL opens up a distinct information pathway. In PLITS, audio tokens are treated identically to text tokens: they are transformed layer by layer through causal self-attention and FFN non–linearities, causing their representations to drift from the original encoder output as they mix with the LLM's internal state. In contrast, LAL uses a dedicated MLP at each layer to project "semantic-ready" audio features directly into the appropriate abstraction for that layer. This preserves a direct link to the audio encoder's semantic output. For tasks that rely on explicit acoustic cues, such as sound event understanding (e.g., *Which animal sound is heard?*), this projection-based injection can be more effective than the deeply transformed representations produced by PLITS.

When comparing to Flamingo, we note that although Flamingo also injects semantic level information without decoding audio tokens inside the LLM, the route by which this information influences text tokens is fundamentally different. In Flamingo, text tokens first attend to audio tokens in a dedicated cross attention module; the resulting signal is added to the text residual stream, and only then do the updated text states interact through the standard self attention layers of the LLM. In LAL, by contrast, audio representations are introduced directly into the same self attention operation as the text tokens, so text attends jointly to audio and text within a single attention computation. This produces a distinct information flow from Flamingo. We also note that LAL does not require the extra cross attention plus FFN adapter blocks used in Flamingo.

To summarize, LAL is similar to PLITS in that it performs in-context injection and allows text tokens to attend over both audio and text tokens, and it is similar to Flamingo in that it injects information that has not been fully decoded inside the LLM. However, it is architecturally distinct from both: LAL is neither a "lite PLITS" nor a "lite Flamingo," but rather a new information pathway for integrating audio encoders with LLMs.

**LAL Integration with Frozen LLM FFN.** We also verify that LAL integration remains effective when the LLM's FFN blocks are frozen, with no significant loss in performance (refer to Appendix E). This finding has important implications for reducing training cost, improving parameter efficiency, and preserving the pretrained knowledge of the LLM while enabling multimodal alignment. For clarity and consistency, however, our main experiments focus on the standard setting with trainable FFN blocks, and discussion of the frozen-FFN variant is limited to Appendix E.

**Leveraging parametric versus contextual knowledge.** Here we posit how LAL *efficiently* utilizes two types of knowledge inherent in pre-trained LLMs: (1) parametric knowledge, primarily embedded within the FFN layers as a result of extensive language pre-training, and (2) contextual knowledge, which is dynamically incorporated through attention mechanisms. We posit that audio as contextual information can effectively induce required concept activations in text token representations via attention-based modulation, without needing direct FFN processing of audio representations. Consequently, audio information indirectly accesses the LLM's parametric knowledge: the audio context "piggybacks" on text tokens, as attention mechanisms reconfigure these representations, which then engage relevant concept-related pathways during FFN processing.

### 3.2.1 COMPUTE AND MEMORY EFFICIENCY.

LAL is more compute- and memory-efficient than PLITS and Flamingo style integration, and the benefits become more pronounced with longer audio sequences. At a high level, the gains come

from reducing the effective attention complexity and avoiding unnecessary routing of audio tokens through the feed-forward sublayers. Quantitative comparisons of memory usage and training throughput are reported in Figure 1.

In the following subsections, we present one-to-one comparisons between LAL and the PLITS baseline when applicable, and otherwise discuss properties specific to LAL.

**Attention Complexity:**

*PLITS:* full causal attention over $N_a + N_t$ tokens with cost $\mathcal{O}\big((N_a+N_t)^2\big)$

*LAL:* only text tokens issue queries; keys and values include audio and text, with cost $\mathcal{O}\big((N_a+N_t)N_t\big)$ eliminating the $N_a^2$ term and all audio to audio interactions.

**Feedforward Routing:**

*PLITS:* audio tokens pass through attention and the feedforward sublayer in every block, increasing floating point operations and activation memory in proportion to $N_a$.

*LAL:* audio tokens do not enter the feedforward sublayer and only serve as keys and values for text queries, which reduces per layer floating point operations and activations stored for backpropagation.

**Scaling With Audio Length:** Non text modalities in multimodal LLMs often yield far more tokens, and audio is no exception. As $N_a$ grows due to longer clips or denser tokenization, PLITS incurs a cost of $(N_a + N_t)^2$, so the $N_a^2$ term dominates. In contrast, LAL scales as $(N_a + N_t)N_t$, which is linear in $N_a$. Thus, the compute and memory gap widens with longer or more finely segmented audio. The feedforward savings in LAL also increase with $N_a$ as a larger share of tokens bypass the most expensive part of each block.

**Distinct from PEFT and LoRA:** LAL is a core architectural modification, not a parameter-efficient fine-tuning (PEFT) method such as LoRA (Hu et al., 2022). Techniques such as LoRA adjust how weights are adapted during training while keeping the forward compute pattern essentially the same at inference. In contrast, LAL changes how audio tokens participate in attention and feedforward routing, so its compute and memory savings apply at inference as well as during training.

### 3.3 PAL: Adding Speech Understanding

Speech occupies a special position among audio modalities because it is closely tied to language and is often described simply as spoken language. In Whisper style systems, speech encoders are trained with transcription style or next text token prediction objectives, so their internal representations form sequences that already resemble linguistic tokens. It is therefore beneficial to use PLITS integration for speech, since this strategy allows the model to decode spoken language in the same space where it already reasons over text and leads to better extraction of speech information.

In contrast, general audio encoders trained with self supervised or contrastive objectives are optimized to produce high level semantic descriptors or event level features rather than language like sequences. For these encoders, it is often sufficient for text tokens to attend to audio features in order to retrieve the relevant information, without requiring the audio tokens themselves to be processed by the LLM feed forward layers. LAL offers a more efficient integration path in this setting because audio tokens appear only as keys and values in attention while the LLM feed forward blocks operate solely on text representations.

This separation is also consistent with classical neuro linguistics: Wernicke's area is primarily associated with comprehension of spoken and written language, while the angular gyrus supports association across auditory, visual, and other sensory inputs. By analogy, speech features may be most effective when interpreted inside a language centric pathway, whereas general audio benefits from a more modality specific route. Empirically, we observe that speech understanding gains from PLITS style decoding inside the LLM for speech encoders such as Whisper. Building an efficient LLM that understands both speech and general audio therefore requires an appropriate allocation of integration strategies between LAL and PLITS. Within our PAL framework, we instantiate this idea in two variants: PAL-MultiEnc (Section 3.3.1), where separate encoders for general audio and for speech are each integrated with the LLM, and PAL-UniEnc (Section 3.3.2), a unified encoder model in which a single audio encoder supports both speech and general audio and interfaces with the LLM.

### 3.3.1 Multi audio encoder PAL

In the multi-encoder PAL architecture, we combine complementary encoders for general audio and speech. General audio encoders such as CLAP and SSLAM provide tokens that capture language aligned semantics and fine grained acoustic detail. These tokens are integrated into the LLM through LAL, so they serve as keys and values in the attention blocks without entering the feed forward pathways.

Speech is handled by a dedicated encoder such as Whisper. The Whisper tokens are mapped into the LLM input space and integrated through the PLITS pathway, where they are prepended to text tokens and processed as full tokens by all LLM layers(Figure 4). This encoder aware allocation allows PAL to use the efficient LAL integration for general audio while reserving compute intensive PLITS integration for speech. We refer to this model as PAL/LAL/PLITS-MultiEnc.

### 3.3.2 Unified audio encoder PAL

In the unified encoder PAL architecture, we use AFWhisper Goel et al. (2025) as a single audio encoder that supports both speech and general audio understanding. AFWhisper produces a sequence of audio tokens for each input clip. To balance efficiency with the benefits of PLITS for speech like content, we construct two parallel views of this sequence.

First, we derive a compact set of summary tokens by applying a one dimensional convolution with stride $r$ along the time axis, which reduces the token count by a factor of $r$. These summary tokens are treated as PLITS tokens: they are mapped into the LLM input space, prepended to the text tokens, and processed as full tokens through all LLM layers.

Second, we retain the complete AFWhisper token sequence for LAL integration. In each attention block, audio information enters as keys and values via these full resolution tokens and the audio summary tokens integrated via PLITS, while queries are issued by text tokens and summary tokens. To preserve temporal ordering, we interleave the tokens in attention(in key and value) so that, within each span of $r$ original AFWhisper tokens, the corresponding summary token is placed after its source tokens (see Figure 5). Concretely, the ordering of keys and values in the attention module follows

$$\mathbf{z} = (\ell_1, \ \ell_2, \ \ldots, \ \ell_r, \ p_1, \ \ell_{r+1}, \ \ldots, \ \ell_{2r}, \ p_2, \ \ldots), \tag{5}$$

where $\ell_i$ denotes an LAL token and $p_j$ denotes a summary token that is integrated via PLITS. This maintains the alignment between summary tokens and their underlying fine grained audio context.

In this way, unified PAL allows the model to benefit from PLITS style decoding over a compact set of audio summaries while still exposing the full AFWhisper token sequence to LAL based attention. We refer to this model as PAL/LAL/PLITS-UniEnc. Additional details such as, visualization(Figure 4), ablations 10 are provided in Appendix E.2.

## 4 Experiments and Results

We empirically evaluate our audio-language framework on a range of audio understanding and reasoning tasks. Unless otherwise specified, we use Llama 3.2 1B Instruct (Grattafiori et al., 2024) as the base LLM. For larger backbones, we report results with Llama 3.2 3B Instruct (Grattafiori et al., 2024), and to assess transfer across model families, we additionally evaluate Qwen2.5 1.5B Instruct (Team, 2024). For audio encoders, we employ SSLAM and CLAP connected via an efficient Q-former-based module that combines their representations without increasing the token count, inspired by Tong et al. (2024); we refer to this connector as **LFST**. We use LFST for all multi encoder experiments unless otherwise specified. In experiments where LFST is not used, we use SSLAM encoder. See Appendix E.1 for further details on **LFST**.

In the following subsections, we present the training setups and results for LAL and PAL.

### 4.1 LAL

**Training Protocol.** We train the proposed audio LLM variants on the one of the largest general audio instruction tuning datasets OpenAQA dataset (Gong et al., 2024) and CompA-R Ghosh et al.

(2024). Our two-stage pipeline comprises: (i) connector pretraining, where only the connector is trained and all other modules are frozen; and (ii) joint training of the connector and the LLM. The audio encoders remain frozen throughout.

For reasoning and open ended question answering we additionally train on open ended data from OpenAQA Gong et al. (2024) as Stage 3 and on the reasoning dataset CompA R Ghosh et al. (2024) as Stage 4. Additional training details are in Appendix C.1.

**Evaluation Protocol.** To assess how effectively LAL transfers critical audio-event information from the encoder to the LLM's latent space, we evaluate on downstream classification, captioning, and reasoning tasks. Following the LTU framework (Gong et al., 2024): (i) for classification, we measure semantic similarity by encoding both model text outputs and target audio labels with `gpt-text-embedding-ada`; (ii) for captioning, we use standard audio captioning datasets and report CIDEr and SPICE.

For reasoning, we adopt the compA-R-test and the evaluation protocol of (Ghosh et al., 2024): we prompt a text-only GPT-4 judge with the audio-LLM's output and auxiliary metadata about the audio events, and obtain scores for *Helpfulness*, *Clarity*, *Correctness*, *Depth*, and *Engagement*. Additional evaluation details are in Appendix D.1.

**Results** To clearly separate contributions, we present two sets of results. First, in Table 1 (classification and captioning) and Table 2 (reasoning), we report a controlled comparison between LAL, Flamingo and PLITS, showing that LAL achieves comparable or better accuracy while being more efficient in speed and memory. Second, in Table 3 (classification and captioning) and Table 4 (reasoning), we compare LAL with prior works. Note that training data scale and model size vary significantly across prior approaches; our model operates on the lower end of both dimensions. These results should therefore be interpreted as evidence that LAL remains competitive despite using fewer resources.

Table 1: Performance evaluation of the proposed efficient integration method **LAL** and SOTA integration **PLITS** across different base LLMs. Evaluation follows the protocol of Gong et al. (2024). FI:Flamingo Integration, AC: Audio caps, CL:Clotho AS2M: AudioSet 2M [†] indicates CIDEr and [‡] indicates SPICE. Other metrics: accuracy (ESC-50, VocalSound), Mi-F1 (DCASE), and mAP (FSD, AudioSet). For evaluation methodology see Section 4.1 and for dataset details see Appendix D

| LLM Backbone | PLITS | FI | LAL | LFST | Classification | | | | | Captioning | | | |
|---|---|---|---|---|---|---|---|---|---|---|---|---|---|
| | | | | | ESC50 | DCASE | VS | FSD | AS2M | AC† | CL† | AC‡ | CL‡ |
| Llama3.2-1B | ✓ | ✗ | ✗ | ✗ | 64.45 | 37.69 | 51.57 | 25.23 | 9.08 | 0.59 | 0.34 | 16.30 | 10.96 |
| | ✗ | ✗ | ✓ | ✗ | 76.70 | 40.97 | **60.87** | 31.44 | 11.83 | 0.66 | 0.38 | 16.97 | 11.87 |
| | ✓ | ✗ | ✗ | ✓ | 84.10 | 45.28 | 57.59 | 42.49 | 14.74 | 0.70 | 0.39 | 17.90 | 11.82 |
| | ✗ | ✓ | ✗ | ✓ | 84.95 | 43.95 | 55.44 | 41.27 | **15.0** | 0.69 | 0.39 | 17.09 | 11.91 |
| | ✗ | ✗ | ✓ | ✓ | **87.40** | **46.23** | 56.03 | **43.91** | 14.74 | **0.72** | **0.42** | **18.08** | **12.58** |
| Llama3.2-3B | ✓ | ✗ | ✗ | ✗ | 70.40 | 40.62 | 61.40 | 28.88 | 10.84 | 0.63 | 0.35 | 16.81 | 11.35 |
| | ✗ | ✗ | ✓ | ✗ | 82.15 | 43.21 | **65.78** | 34.29 | 12.91 | 0.67 | 0.38 | 17.80 | 12.18 |
| | ✓ | ✗ | ✗ | ✓ | 84.60 | 46.16 | 59.15 | 43.29 | 15.00 | 0.7 | 0.38 | 17.9 | 12.03 |
| | ✗ | ✗ | ✓ | ✓ | **89.25** | **47.21** | 60.46 | **43.86** | **15.03** | **0.73** | **0.40** | **18.61** | **12.46** |
| Qwen2.5-1.5B | ✓ | ✗ | ✗ | ✗ | 68.00 | 37.57 | 56.45 | 27.87 | 9.56 | 0.63 | 0.38 | 16.63 | 11.74 |
| | ✗ | ✗ | ✓ | ✗ | 70.85 | 38.79 | **59.20** | 28.53 | 10.28 | 0.63 | 0.38 | 16.65 | 11.44 |
| | ✗ | ✗ | ✓ | ✓ | **87.80** | **45.52** | 56.73 | **43.26** | 13.92 | **0.73** | **0.41** | **18.45** | **12.20** |

Table 2: GPT-4 evaluation of LAL and PLITS on the CompA-R benchmark (Ghosh et al., 2024). A text only GPT-4 judge scores the model outputs; see Ghosh et al. (2024) for the detailed prompt.

| PLITS | LAL | LFST | Helpfulness | Clarity | Correctness | Depth | Engagement |
|---|---|---|---|---|---|---|---|
| ✓ | ✗ | ✓ | **3.86** | **4.74** | **3.84** | 2.86 | 2.99 |
| ✗ | ✓ | ✓ | 3.85 | 4.70 | 3.82 | **2.88** | **3.01** |

## 4.2 PAL

**Training Protocol.** PAL follows the same two stage procedure as LAL: (i) connector pretraining, where only the connector is trained and all other modules are frozen; and (ii) joint training of the

Table 3: Comparison of LAL classification and captioning performance with prior works. Except for Audio Flamingo 2, all other systems use PLITS; their higher scores mainly stem from larger datasets, bigger LLMs, and stronger audio encoders.

| Models | Classification | | | | | Captioning | | | |
|---|---|---|---|---|---|---|---|---|---|
| | ESC50 | DCASE | VS | FSD | AS2M | AC† | CL† | AC‡ | CL‡ |
| Pengi-124M | **91.9** | 33.8 | 60.3 | 46.7 | - | - | - | - | - |
| SALMONN-7B | 16.4 | 18.0 | 16.9 | 22.1 | 13.4 | - | - | 8.3 | 7.6 |
| Audio Flamingo-2-3B | 83.9 | - | - | **47.9** | - | 0.58 | **0.46** | - | - |
| LTU-7B | 83.1 | 45.9 | 55.6 | 46.3 | 18.7 | - | - | 17 | 11.9 |
| GAMA-7B | 82.6 | 38.4 | 52.4 | 47.8 | **19.2** | - | - | 18.5 | 13.5 |
| LAL-1B (Ours) | 87.40 | 46.23 | 56.03 | 43.91 | 14.74 | 0.72 | 0.42 | 18.08 | 12.58 |
| LAL-3B (Ours) | 89.25 | **47.21** | **60.46** | 43.86 | 15.03 | **0.73** | 0.40 | **18.61** | 12.46 |

Table 4: LAL performance comparison with prior works for the reasoning (CompA-R) task. All prior works use PLITS integration. Their higher scores mainly stem from larger datasets, bigger LLMs, and stronger audio encoders.

| Models | Clarity | Correctness | Engagement | Avg |
|---|---|---|---|---|
| Qwen-Audio-Chat-8B (Chu et al., 2023) | 3.5 | 3.3 | 3.6 | 3.5 |
| LTU-7B (Gong et al., 2024) | 3.5 | 3.2 | 3.4 | 3.4 |
| SALMONN-7B (Tang et al., 2024) | 2.6 | 2.4 | 2.0 | 2.3 |
| Pengi-124M (Deshmukh et al., 2023) | 1.8 | 1.5 | 1.3 | 1.5 |
| LTU w/ CompA-R-7B (Gong et al., 2024) | 3.5 | 3.2 | 3.4 | 3.6 |
| GAMA-IT-7B (Ghosh et al., 2024) | 4.3 | **3.9** | **3.9** | **4.0** |
| LAL-1B (Ours) | **4.70** | 3.82 | 3.01 | 3.80 |

connector and the LLM. The audio encoders remain frozen throughout. For Stage 1, we construct a mixture from the general audio OpenAQA Stage 1 set, augmented with the OpenASQA (Gong et al., 2023b) Stage 1 split for speech understanding. For Stage 2, we use a curated audio, speech, and music reasoning instruction tuning corpus, specifically a 6M subset of AudioSkills (Goel et al., 2025).

**Evaluation Protocol.** We first target speech understanding with two tasks: speech recognition and speaker gender classification (using `gpt-text-embedding-ada` as explained in Section 4.1); We then assess general audio, music, and speech reasoning on MMAR Ma et al. (2025), MMAU Sakshi et al. (2024) and MMSU Wang et al. (2025) which report detailed category wise performance.

**Results.** Our experiments on the speech understanding and reasoning benchmark MMSU(Table 12) (refer to Appendix E.3), speech based emotion recognition, gender classification (Table 11) and the speech subsets in MMAU (Table 5) and MMAR (Table 6) show that LAL consistently exhibits reduced performance compared to PLITS on speech based tasks. This substantiates the need for the hybrid PAL architecture.

From our evaluation results in Table 11 for classification and Tables 5 and 6 for reasoning, PAL is comparable to PLITS in accuracy while retaining efficiency advantages in both multi encoder and unified encoder setups. In the multi encoder setup, we also observe that adding a Whisper encoder changes performance in the general audio (sound) and music domains. We hypothesize that this is because Whisper encodes background sounds, as reported by Gong et al. (2023a), which provides some event detection capability.

Our PAL versus PLITS comparison is controlled within our setup, using the same backbone, data, and training hyperparameters; see Appendix C.2 for details. The primary comparison in these tables is therefore between PAL, LAL and PLITS, and results from prior work are included only to place PAL in the broader literature. With the exception of Audio Flamingo 2, the other systems are based on PLITS. The higher scores reported by some prior systems over our PLITS baseline largely reflect larger training sets, larger LLMs, and stronger audio encoders. This work assesses the integration in isolation, which is why we focus on the PAL versus PLITS comparison.

Table 5: Evaluation on **MMAU-v05.15.25** (Sakshi et al., 2024) (accuracy, %). Sound (Sn), Music (Mu), Speech (Sp), and r (reduction factor, Section 3.3.2). Except for Audio Flamingo 2, all other systems use PLITS; their higher scores mainly stem from larger datasets, bigger LLMs, and stronger audio encoders. **Boldface** marks PAL multi encoder and unified encoder variants separately, reflecting our focus on integration.

| Model | Sn | | Mu | | Sp | | Total (Avg) | |
|---|---|---|---|---|---|---|---|---|
| | mini | test | mini | test | mini | test | mini | test |
| Step-Audio-2-mini-8.3B (Wu et al., 2025a) | 79.30 | 75.57 | 68.44 | 66.85 | 66.18 | 66.49 | 72.73 | 70.23 |
| DeSTA2.5-Audio-8B (Lu et al., 2025) | 70.27 | 66.83 | 56.29 | 57.10 | 71.47 | 71.94 | 66.00 | 65.21 |
| SALMONN-13B (Tang et al., 2024) | 41.14 | 42.10 | 37.13 | 37.83 | 26.43 | 28.77 | 34.90 | 36.23 |
| GAMA-7B (Ghosh et al., 2024) | 31.83 | 30.73 | 17.71 | 17.33 | 12.91 | 16.97 | 20.82 | 21.68 |
| GAMA-IT-7B (Ghosh et al., 2024) | 30.93 | 32.73 | 26.74 | 22.37 | 10.81 | 11.57 | 22.83 | 22.22 |
| LTU-7B (Gong et al., 2024) | 20.42 | 20.67 | 15.97 | 15.68 | 15.92 | 15.33 | 17.44 | 17.23 |
| Qwen2.5-Omni-7B (Xu et al., 2025) | 78.10 | 76.77 | 65.90 | 67.33 | 70.60 | 68.90 | 71.50 | 71.00 |
| Qwen2-Audio-Instruct-7B (Chu et al., 2024) | 67.27 | 61.17 | 56.29 | 56.29 | 55.67 | 55.57 | 59.90 | 57.40 |
| M2UGen-7B (Liu et al., 2024b) | 43.24 | 42.44 | 37.13 | 38.53 | 35.37 | 35.77 | 37.90 | 39.76 |
| MusiLingo-7B (Deng et al., 2024) | 43.24 | 41.93 | 40.12 | 41.23 | 31.23 | 31.73 | 38.10 | 38.29 |
| Audio Flamingo-3-8.2B (Goel et al., 2025) | 79.58 | 75.83 | 73.95 | 74.47 | 66.37 | 66.97 | 73.30 | 72.42 |
| Audio Flamingo-2-3B (Ghosh et al., 2025a) | 71.47 | 68.13 | 70.96 | 70.20 | 44.74 | 44.87 | 62.40 | 61.06 |
| Audio Flamingo Chat-1B (Kong et al., 2024) | 25.3 | 23.33 | 17.66 | 15.77 | 6.91 | 7.67 | 16.60 | 15.59 |
| PLITS-MultiEnc-1B (Baseline) | 71.17 | **72.20** | 71.56 | 69.66 | 53.45 | 54.31 | 65.40 | **64.61** |
| LAL-MultiEnc-1B (Ours) | 71.77 | 70.39 | 70.96 | 66.50 | 45.65 | 48.17 | 62.80 | 61.85 |
| PAL-MultiEnc-1B (Ours) | **72.07** | 70.63 | 70.66 | 66.10 | **53.45** | 53.28 | **65.40** | 63.45 |
| PLITS-UniEnc-3B (Baseline) | 75.68 | 72.03 | **70.96** | 69.63 | 46.25 | 46.48 | 64.30 | 62.91 |
| PAL-UniEnc-3B(r=3)(Ours) | **76.28** | **73.87** | 69.76 | **70.03** | 49.25 | 54.46 | 65.10 | **66.26** |

Table 6: Evaluation of PAL on **MMAR** (Ma et al., 2025) (accuracy, %). Abbr: Sound (Sn), Music (Mu), Speech (Sp) and r (reduction factor, Section 3.3.2). Except for Audio Flamingo 2, all other systems use PLITS; their higher scores mainly stem from larger datasets, bigger LLMs, and stronger audio encoders. **Boldface** marks PAL multi encoder and unified encoder variants separately, reflecting our focus on integration.

| Models | Sn | Mu | Sp | Mix Sn-Mu | Mix Sd-Sp | Mix Mu-Sp | Mix Sn-Mu-Sp | Total Accuracy |
|---|---|---|---|---|---|---|---|---|
| Audio Flamingo-2-3B | 24.85 | 17.48 | 20.75 | 18.18 | 26.61 | 23.17 | 8.33 | 21.90 |
| Audio Flamingo-3-8.2B | - | - | - | - | - | - | - | 58.5 |
| LTU-7B | 19.39 | 19.90 | 13.95 | 18.18 | 24.77 | 21.95 | 16.67 | 19.20 |
| SALMONN-13B | 30.30 | 31.07 | 34.69 | 9.09 | 34.86 | 35.37 | 41.67 | 33.20 |
| GAMA-7B | 29.09 | 24.27 | 27.89 | 27.27 | 24.77 | 28.05 | 20.83 | 26.50 |
| GAMA-IT-7B | 22.42 | 16.02 | 12.24 | 36.36 | 22.48 | 14.63 | 12.50 | 17.40 |
| Qwen2.5-Omni-7B | 58.79 | 40.78 | 59.86 | 54.55 | 61.93 | 67.07 | 58.33 | 56.70 |
| PLITS-MultiEnc-1B(Baseline) | 38.79 | **42.72** | **40.48** | 18.18 | 44.50 | 39.02 | **41.67** | 41.20 |
| LAL-MultiEnc-1B(Ours) | 40.00 | 40.29 | 35.71 | 27.27 | 42.66 | 43.90 | 37.50 | 39.50 |
| PAL-MultiEnc-1B(Ours) | **40.61** | 41.75 | 38.10 | **36.36** | **45.87** | **52.44** | **41.67** | **42.20** |
| PLITS-UniEnc-3B(Baseline) | 38.79 | 40.29 | 37.41 | **36.36** | **48.17** | 40.24 | **50.00** | 41.10 |
| PAL-UniEnc-3B(r=3)(Ours) | **46.61** | **44.17** | **40.82** | 27.27 | **48.17** | 46.34 | 41.67 | **44.40** |

## 5 CONCLUSION

We introduce LAL, which injects audio only through attention keys and values and skips feedforward processing for audio tokens. This reduces attention interactions and activations, yielding up to about 60% lower memory usage and up to about 190% higher training throughput, with performance comparable to PLITS, the state of the art baseline integration for classification, captioning, and reasoning tasks. We also propose PAL, an hybrid integration that uses LAL both PLITS for efficient audio-LLM that understand general audio and speech. LAL is a core architectural change rather than a parameter efficient fine tuning method, so the efficiency gains hold at inference and during training. For future work, we plan to scale to larger backbones, use higher quality instruction data to improve reasoning, and explore streaming and long context audio.

ETHICS STATEMENT

All experiments use publicly available datasets. The proposed approach enables beneficial applications, but it could also be misused, for example to monitor individuals without consent. We acknowledge these risks and will release code and models with care, including clear documentation and use guidance to support responsible research.

REPRODUCIBILITY STATEMENT

Implementation details are provided in Sections 3.2 and 3.3. Training details appear in Appendix C, and the evaluation protocol is described in Appendix D. Code and pretrained models will be made available upon acceptance.

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

# A   APPENDIX

# B   LLM USAGE

Large language models were used only as assistive tools for editing and polishing text. We followed the benchmark protocol of Ghosh et al. (2024) to rate audio LLM outputs; the GPT based evaluation is part of that benchmark. See Section 4.1 for details. LLMs were not used for model design, data selection, experiment setup, implementation, analysis, or generation of results. All technical content was written and verified by the authors.

# C   TRAINING DETAILS

## C.1   LAL TRAINING DETAILS

We use OpenAQA (Gong et al., 2024) two stage training setup for LAL to report the results in Table 1. We also train on broader open ended data from OpenAQA (Gong et al., 2024) and on the reasoning dataset CompA R (Ghosh et al., 2024), with evaluations shown in Table 2. Additional training hyperparameters appear in Table 7.

## C.2   PAL TRAINING DETAILS

PAL uses a two stage training protocol(Table 8). In Stage 1, we start from the Stage 1 dataset used for LAL and augment it with additional speech focused data from OpenASQA (Gong et al., 2023b). In Stage 2, we fine tune on a curated audio, speech, and music reasoning instruction corpus, AudioSkills (Goel et al., 2025). We use a 6M example subset of AudioSkills (from the original 10M) due to the unavailability of original audio files for some source datasets.

Table 7: Hyper-parameters used for the three stage training of **LAL** and **PLITS** (Llama3.2 1B)

| Training Configuration | Stage 1 (Connector Pre training) | Stage 2 (LLM Fine tuning) | Stage 3 \| Stage 4 (LLM Fine tuning) |
|---|---|---|---|
| Optimizer | AdamW (Loshchilov & Hutter, 2017) | | |
| Learning Rate Schedule | Cosine (Loshchilov & Hutter, 2016) | | |
| Peak Learning Rate | 0.001 | 0.0001 | 0.0001 |
| Epochs | 1 | 1 | 1 |
| Warm up Ratio (steps) | 0.05 | 0.03 | 0.03 |
| Dataset Size | 1.2 M | 1.9 M | 5.6 M \| 200 K |
| Batch Size | 32 | 12 | 12 |
| Gradient Accumulation Steps | 4 | | |
| GPUs | 2× Nvidia A100 (80GB) | | |
| RAM | 150 GB | | |
| Loss | Next token loss on text part | | |

Table 8: Hyperparameters used for the two stage training of **PAL** and **PLITS** (Llama3.2 1B)

| Training Configuration | Stage 1 (Connector Pre training) | Stage 2 (LLM Fine tuning) |
|---|---|---|
| Optimizer | AdamW (Loshchilov & Hutter, 2017) | |
| Learning Rate Schedule | Cosine (Loshchilov & Hutter, 2016) | |
| Peak Learning Rate | 0.001 | 0.0001 |
| Epochs | 1 | 1 |
| Warm up Ratio (steps) | 0.05 | 0.03 |
| Dataset Size | 1.7 M | 6.4 M |
| Batch Size | 16 | 4 |
| Gradient Accumulation Steps | 2 | 32 |
| GPUs | 4× Nvidia A100 (64GB) | |
| RAM | 250 GB | |
| Loss | Next token loss on text part | |

# D    EVALUATION DETAILS

## D.1    LAL EVALUATION DETAILS

We follow the evaluation protocol of Gong et al. (2024) for classification and captioning, and use the CompA R test set of Ghosh et al. (2024) for reasoning. Below we summarize the datasets included in the Gong et al. (2024) protocol.

**VocalSound** (Gong et al., 2022b): The VocalSound dataset consists of 21,024 crowd-sourced recordings of 6 different classes of vocal expressions collected from 3,365 unique subjects. We evaluated our model on the VocalSound evaluation set which contains 3,594 audio clips, and report top-1 accuracy scores across the 6 classes for single-class classification performance. It is important to note that VocalSound was excluded from our training data; therefore, our evaluation on Vocal-Sound is considered zero-shot.

**ESC-50** (Piczak, 2015): The ESC-50 dataset comprises 2,000 five-second environmental audio clips categorized into 50 different classes. Following Gong et al. (2024), we evaluate our model on all 2,000 audio samples and report the top-1 accuracy score for single-class classification performance. It is important to note that while ESC-50 is originally sampled from the Freesound dataset (which is included in our training data), ESC-50 itself was excluded from training. Therefore, our evaluation on this dataset is considered a weak zero-shot evaluation.

**DCASE2017 task 4** (DCASE) (Mesaros et al., 2019): DCASE 2017 Task 4 contains 17 sound events distributed across two categories: "Warning" and "Vehicle". The evaluation set consists of 1,350 audio clips. We evaluated our model on this dataset and report micro F1-score(MiF1) for single-class classification performance. It is important to note that DCASE 2017 task 4 is originally sampled from AudioSet, which is included in our training data. However, DCASE 2017 task 4 itself is excluded from training, making our evaluation on this dataset a weak zero-shot evaluation.

**FSD50K** (FSD) (Fonseca et al., 2021): The FSD50K evaluation set contains 10,231 audio clips. We evaluated our model on this evaluation set and report the mAP score for multi-label classification performance. Since the training and validation sets of FSD50K are included in our training data, this evaluation is considered an in-domain evaluation.

**AudioSet** (Gemmeke et al., 2017): We evaluated our model on this evaluation set and report the mAP score for multi-label classification performance. The training set of AudioSet is included in our training data, making this evaluation an in-domain evaluation.

**AudioCaps** (Kim et al., 2019): The AudioCaps evaluation set contains 901 audio clips, each paired with 5 audio captions, resulting in a total of 4,505 audio-caption pairs. We evaluated our model on this evaluation set and report the captioning scores using CIDER and SPICE metrics. The training and validation sets of AudioCaps are included in our training data, making this evaluation an in-domain evaluation.

**Clotho V2** (Drossos et al., 2020): The Clotho V2 evaluation set contains 1,045 audio clips, each paired with 5 audio captions, resulting in a total of 5,225 audio-caption pairs. We evaluated our model on this evaluation set and report the captioning scores using CIDER and SPICE metrics. The development and validation sets of Clotho V2 are included in our training data, making this evaluation an in-domain evaluation.

### D.2 PAL EVALUATION DETAILS

For speech classification (emotion recognition and gender classification), we follow the protocol of Gong et al. (2023b). For combined sound, speech, and music reasoning, we evaluate on the standard benchmark datasets MMAU (Sakshi et al., 2024) and MMAR (Ma et al., 2025).

## E LAL INTEGRATION WITH FROZEN LLM FFN

Standard audio-LLM training typically requires full fine tuning of the LLM. However, since LAL integrates audio information solely through the attention mechanism, we investigate whether LAL remains effective when the LLM feedforward (FFN) blocks, which are widely believed to encode much of the model's factual and linguistic knowledge, are frozen and only the attention layers are updated. In Stage 2 of our training pipeline, we therefore construct a variant with the LLM FFN frozen. As shown in Table 9, performance is largely maintained under this setting. This result suggests that LAL can successfully integrate audio information through attention without modifying the knowledge stored in the FFN modules. Such a property has important implications for reducing training cost, improving parameter efficiency, and preserving the pretrained knowledge of the LLM while enabling multimodal alignment.

Table 9: Performance evaluation of the **LAL** Integration with frozen FFN. Evaluation follows the protocol of Gong et al. (2024). AC: Audio caps, CL:Clotho AS2M: AudioSet 2M [†] indicates CIDEr and [‡] indicates SPICE. Metrics: accuracy (ESC-50, VocalSound), Mi-F1 (DCASE), and mAP (FSD, AudioSet). Complete evaluation methodology explained in Section 4.1 and dataset details in Appendix D

| LLM Backbone | FFN Frozen | PLITS | LAL | LFST | Classification | | | | | Captioning | | | |
|---|---|---|---|---|---|---|---|---|---|---|---|---|---|
| | | | | | ESC50 | DCASE | VS | FSD | AS2M | AC[†] | CL[†] | AC[‡] | CL[‡] |
| Llama3.2-1B | ✗ | ✓ | ✗ | ✗ | 64.45 | 37.69 | 51.57 | 25.23 | 9.08 | 0.59 | 0.34 | 16.30 | 10.96 |
| | ✗ | ✗ | ✓ | ✗ | **76.70** | **40.97** | **60.87** | **31.44** | **11.83** | **0.66** | 0.38 | **16.97** | **11.87** |
| | ✓ | ✗ | ✓ | ✗ | 71.80 | 33.99 | 55.28 | 29.38 | 10.48 | 0.63 | **0.40** | 16.11 | 11.75 |

### E.1 LFST CONNECTOR: LANGUAGE ALIGNED AND FINE GRAINED SPATIOTEMPORAL CONNECTOR

We adopt the connector *proposed in Cambrian* (Tong et al., 2024) and apply it in our audio setting to fuse a language aligned encoder such as CLAP with a self supervised encoder such as SSLAM. The connector produces a compact set of latent tokens that combine semantic cues from CLAP with

fine grained spatiotemporal detail from SSLAM, while keeping sequence length fixed and avoiding the overhead of naive concatenation.

**Formalization.** Let the encoder outputs be

$$H_{\text{sslam}},\ H_{\text{clap}} \in \mathbb{R}^{F \times T \times d}, \quad z \in \mathbb{R}^d,$$

where $F$ is frequency, $T$ is time, and $d$ is the feature dimension. Following Tong et al. (2024), a single latent token $z$ is broadcast to each spatiotemporal location, yielding $z_{f,t}$ for every $(f,t)$. Inside the connector, which consists of 3 cross attention layers, each $z_{f,t}$ is updated through cross attention with the corresponding local regions of $H_{\text{sslam}}$ and $H_{\text{clap}}$. To preserve temporal structure when flattening across $(F,T)$, we insert a *newline token* along the frequency axis so that each new time step begins with this marker before its spectral tokens (see Figure 3).

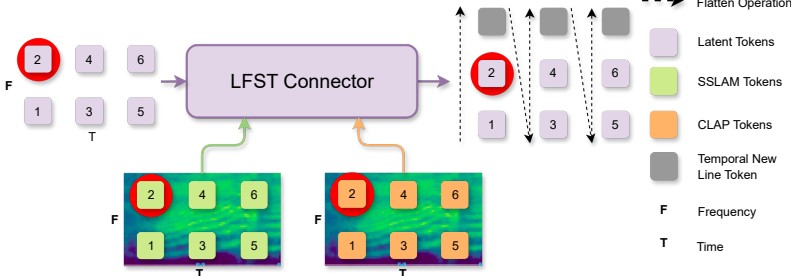

Figure 3: Overview of LFST using the Cambrian connector (Tong et al., 2024). A single latent token is broadcast to every time–frequency location and then updated inside the connector by cross attention with local SSLAM and CLAP features, fusing fine grained spatiotemporal detail with language aligned semantics. The red tokens illustrate the latent query and the local encoder keys and values it attends to. A newline token is inserted at each new time step so the flattened sequence preserves the original spatiotemporal layout while keeping the output length fixed.

## E.2 PAL 2 Variants: Multi Audio Encoder and Unified Audio Encoder Visualizations

This section provides visualizations of the two PAL variants discussed in Section 3.3.1 and Section 3.3.2. The multi-encoder audio configuration is illustrated in Figure 4, while the unified audio encoder configuration is presented in Figure 5.

Table 10: Performance metrics for PLITS and PAL unified encoder variants. Throughput (Samples/s) and memory (VRAM) are measured during training and inference. Evaluation metrics (MMAR, MMAU, MMSU) represent average performance across benchmark tasks. r (reduction factor, Section 3.3.2). (↑ = higher is better, ↓ = lower is better).

| Model | Training/Inference | | Evaluation Performance | | |
|---|---|---|---|---|---|
| | Samples/s ↑ | VRAM (GB) ↓ | MMAR ↑ | MMAU ↑ | MMSU ↑ |
| PLITS-UniEnc-3B | 70.68/7.80 | 42.49/17.68 | 41.10 | 62.91 | 40.12 |
| PAL-UniEnc-3B(r=3) | 96.12/8.40 | 41.48/12.99 | **44.40** | **66.26** | **43.44** |
| PAL-UniEnc-3B(r=5) | **105.72/8.76** | **39.98/11.71** | 42.00 | 63.42 | 41.22 |

## E.3 LAL vs. PLITS Integration for Speech

In this section, we provide a detailed analysis of different integration strategies for the speech modality. While LAL demonstrates high efficiency and strong performance for general audio events, our experiments indicate that speech understanding, which requires decoding linguistic content, benefits significantly from the PLITS integration strategy. This observation motivates our hybrid PAL architecture.

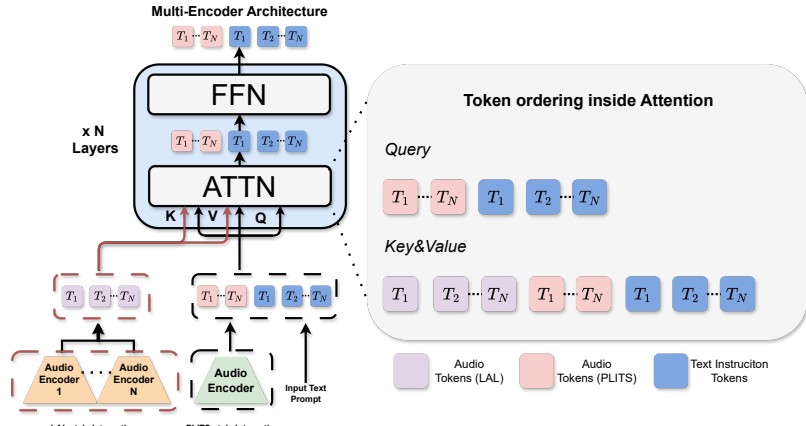

Figure 4: Overview of Multi-Encoder Architecture. Multiple audio encoders process input audio in parallel, with each encoder producing audio tokens. Purple tokens in the diagram represent audio tokens that follow LAL integration $[T_1^{LAL}, \ldots, T_N^{LAL}]$, while red tokens represent audio tokens that follow PLITS integration $[T_1^{PLITS}, \ldots, T_N^{PLITS}]$. Blue tokens represent text instruction tokens $[T_1^{text}, \ldots, T_N^{text}]$. In the attention mechanism, the query is ordered as $[T^{PLITS}, T^{text}]$. The key and value tensors are ordered as $[T^{LAL}, T^{PLITS}, T^{text}]$.

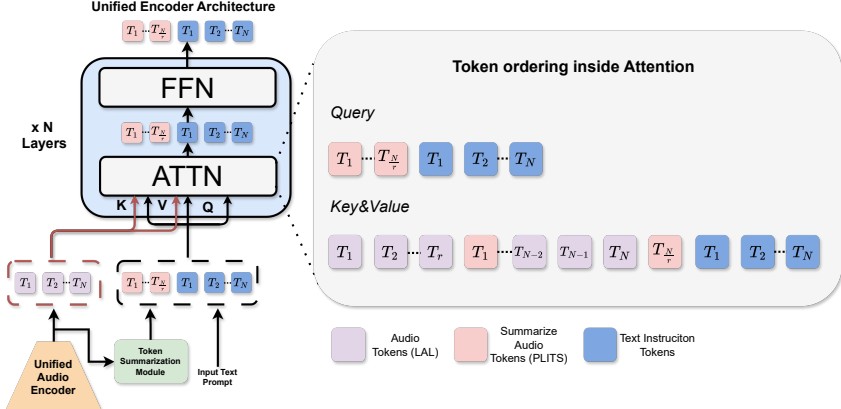

Figure 5: Overview of PAL integration with unified audio encoder. The unified audio encoder processes input audio and produces audio tokens that are split into two paths. Tokens following LAL integration are $[T_1^{LAL}, \ldots, T_N^{LAL}]$, while token summarization reduces them by factor $r$ for PLITS integration producing $[T_1^{PLITS}, \ldots, T_{N/r}^{PLITS}]$. Purple tokens represent LAL audio tokens, red tokens represent summarized PLITS audio tokens, and blue tokens represent text tokens $[T_1^{text}, \ldots, T_N^{text}]$. In the attention mechanism, the query is ordered as $[T^{PLITS}, T^{text}]$ from PLITS integration, while the key and value tensors are ordered as the interleaving of audio tokens $T^{LAL}$ with their corresponding summarized tokens $T^{PLITS}$, then text tokens $T^{text}$ (Section 3.3.2).

**Speech Classification (IEMOCAP & VoxCeleb2).** We evaluate on speech-specific classification tasks: emotion recognition (IEMOCAP) and gender classification (VoxCeleb2). Table 11 compares three configurations: pure PLITS (both encoders use PLITS), pure LAL (both use LAL), and the hybrid PAL configuration (general audio via LAL, speech via PLITS). While the pure LAL configuration performs comparably to pure PLITS on these classification tasks, the hybrid PAL configuration yields the highest accuracy on both datasets (68.81% on IEMOCAP and 97.99% on VoxCeleb2).

**Speech Understanding and Reasoning (MMSU).** Further evaluation on speech understanding we benchmark using the MMSU: A Massive Multi-task Spoken Language Understanding and Reasoning Benchmark(MMSU) (Wang et al., 2025). Table 12 presents the performance of Multi-Encoder models on the MMSU, which assesses both paralinguistic and linguistic capabilities. We observe a clear performance gap between the pure PLITS and pure LAL baselines: the LAL-MultiEnc model under performs the PLITS-MultiEnc baseline. The PAL architecture, which routes speech through PLITS and general audio through LAL, recovers this performance loss.

These results substantiate our architectural choice for PAL: treating speech as a "language-like" modality that requires deeper integration via PLITS, while treating general audio as "contextual" information that is well suited for the lightweight LAL integration.

Table 11: Integration choices for Whisper and CLAP/SSALM on multiple audio encoder setting (Section 3.3.1) evaluated on IEMOCAP (Busso et al., 2008) (emotion recog.) and Vox-Celeb2 (Hechmi et al., 2021) (gender cls.) (accuracy, %).

| SSLAM+CLAP Integration | Whisper Integration | IEMOCAP | Voxceleb2 |
|---|---|---|---|
| PLITS | PLITS | 65.67 | 96.69 |
| LAL | LAL | 66.88 | 97.19 |
| LAL | PLITS | **68.81** | **97.99** |

Table 12: Evaluation of PLITS, LAL, and PAL on **MMSU** (accuracy). Abbr: P-Per = Paralinguistic Perception, L-Per = Linguistic Perception, L-Res = Linguistic Reasoning, P-Res = Paralinguistic Reasoning, Per = Perception (avg), Res = Reasoning (avg), r (reduction factor, Section 3.3.2)

| Model | MMSU | | | | | | |
|---|---|---|---|---|---|---|---|
| | P-Per | L-Per | L-Res | P-Res | Per | Res | Overall |
| PLITS-MultiEnc-1B | **33.86** | **33.69** | 58.47 | 45.67 | **33.76** | 56.69 | **44.86** |
| LAL-MultiEnc-1B | 33.56 | 30.32 | 51.08 | **46.27** | 31.59 | 50.41 | 40.70 |
| PAL-MultiEnc-1B | 32.67 | 29.62 | **58.66** | 45.97 | 30.81 | **56.90** | 43.44 |
| PLITS-UniEnc-3B | 34.75 | 28.79 | 50.79 | 42.99 | 31.12 | 49.71 | 40.12 |
| PAL-UniEnc-3B(r=3) | **37.92** | **30.70** | 54.87 | **47.16** | 33.53 | 53.80 | **43.34** |

### E.4 LAL: PRESERVATION OF TOKEN ORDER INFORMATION

In the standard PLITS integration paradigm, audio tokens are mapped into the LLM input space and physically prepended (or inserted) into the text token sequence. Consequently, the model assigns sequential position IDs across the entire concatenated sequence-for example, $[1, \ldots, N_{sys}]$ for the system prompt, $[N_{sys}+1, \ldots, N_{sys}+N_{audio}]$ for the audio tokens, and $[N_{sys}+N_{audio}+1, \ldots, N_{total}]$ for the user prompt. These position IDs are used by the Rotary Positional Embeddings (RoPE) in the query (Q) and key (K) projections to encode relative and absolute positions, which is crucial for the attention mechanism to function correctly.

In our LAL implementation, audio tokens are not part of the LLM's input text sequence but are injected directly into the attention mechanism as keys and values. To ensure that the model retains accurate temporal ordering and relative distance information, we explicitly manage the position IDs to mirror the structure of PLITS.

We implement this by adjusting the position IDs of the text tokens to leave a *gap* corresponding to the length of the audio sequence. Specifically, if the system prompt occupies indices $[1, \ldots, k]$, we do not assign the immediate next integer to the user prompt. Instead, we shift the starting position ID of the user prompt to $k + N_{audio} + 1$, effectively reserving the interval $[k+1, \ldots, k+N_{audio}]$ for the audio tokens. Inside the attention module, we assign these reserved position IDs to the audio keys and values as illustrated in Figure 6.

Crucially, this adjustment ensures that for every text token, the position ID used for its Query representation is identical to the position ID used for its corresponding Key and Value representations. By maintaining this consistency, the model preserves the correct self-attention structure for text while integrating audio context at the appropriate relative positions. We apply equivalent position ID adjustments in both the Multi-Encoder and Unified-Encoder variants of PAL to maintain token order integrity across all architectures.

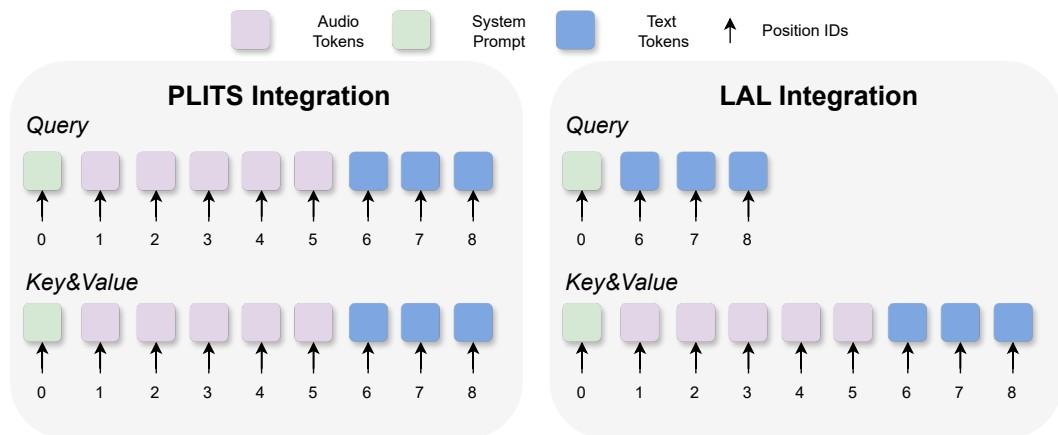

Figure 6: LAL: Preservation of Token Order Information. This diagram illustrates how LAL preserves temporal ordering when integrating audio tokens into the LLM's attention mechanism. LAL manages position IDs by creating a gap for the audio sequence. By shifting the user prompt's position ID to $k + N_{audio} + 1$, LAL reserves the interval $[k+1, \ldots, k+N_{audio}]$ for audio tokens, ensuring that text token Query, Key, and Value representations maintain identical position IDs and preserve correct self-attention structure.

### E.5 EXTENDED LITERATURE REVIEW

**Audio Representation learning** To obtain rich semantic audio representations, recent advances in audio representation learning have led to powerful audio encoders trained with diverse objectives across different pretraining paradigms. The studies have shifted from simple supervised learning paradigms (Gong et al., 2022a; 2021) to more complex self-supervised paradigms (Huang et al., 2022; Ahmed et al., 2024; Chen et al., 2024; Alex et al., 2025) that employ contrastive objectives and masked-token prediction strategies to capture both global semantic structure and fine-grained local details within audio representations. Furthermore, in the multimodal pretraining paradigm, language-aligned audio representations are obtained through contrastive audio–language models (Elizalde et al., 2023; Wu et al., 2023; Ghosh et al., 2025b) which align the representations of audio and language into a unified semantic space. Transcription-based approaches (Radford et al., 2023) leverage next token prediction on speech-to-text tasks to learn robust audio representations that capture speech semantics and acoustic-linguistic relationships.

