# OpenReview forum: "PAL: Probing Audio Encoders via LLMs - Audio Information Transfer into LLMs"
_ICLR.cc/2026/Conference — Submitted to ICLR 2026_

### Official Review · Reviewer_qmDt · 2025-10-29

**Soundness:** 3
**Presentation:** 2
**Contribution:** 2
**Rating:** 4
**Confidence:** 2

**Summary:**

This paper aims to improve the efficiency of information transfer from audio encoders to large language models (LLMs). It introduces LAL, which injects audio representations through attention mechanisms only, and PAL, which integrates PLITS and LAL for encoder-aware fusion.

**Strengths:**

1. Through architectural redesign, LAL improves both training and inference efficiency.
2. LAL demonstrates high computational efficiency compared to PLITS baselines without performance degradation, indicating potential benefits for future multimodal LLMs.

**Weaknesses:**

1. The advantage of LAL over existing Flamingo-style architectures is not fully convincing, as the paper lacks efficiency comparisons with Flamingo-style baselines. If my understanding is correct, the claimed improvement over Flamingo-style architectures lies in bypassing the audio FFNs to improve efficiency. However, since the main computational bottleneck in PLITS arises from audio self-attention ($O(N_a^2)$), it is unclear whether modifying the audio FFNs meaningfully contributes to reducing the overall time complexity by another $O(N_a)$. Moreover, Figure 1 only presents comparisons against PLITS baselines, without including Flamingo-style counterparts.
2. The efficiency of PAL is not reported. Figure 1 only shows comparisons between LAL and PLITS; how PAL performs relative to PLITS baselines remains unclear.
3. The criterion for selecting between LAL and PLITS in PAL is ambiguous. When the audio encoder differs from those used in the paper, it is unclear which variant (LAL or PLITS) would be more suitable or likely to perform better.
4. The writing could be improved. Section 3 (*Methodology*) mixes method and results, which should be clearly separated. In addition, the structure of PAL in Figure 2(C) is not fully explained in the text and lacks proper legends.

**Questions:**

Could the authors please address:

1. How efficient are LAL and PAL compared with Flamingo-style architectures?
2. Is PAL a new architecture or simply a selection mechanism that switches between LAL and PLITS depending on the encoder? In Figure 2, PAL’s structure appears different from both, yet Section 3.4 states that PAL chooses the integration per encoder between LAL and PLITS, which contradicts Figure 2. If this interpretation is correct, does it mean that when the encoder is Whisper, PAL simply reduces to PLITS?
3. In Figure 2, what do the different input square colors (purple, pink, and blue) represent?
4. In Tables 6 and 7, which models are considered fair comparisons? Some baselines outperform PAL but are not boldfaced. Does this mean they are not comparable, and if so, why? For example, Audio Flamingo-2-3B is listed in Tables 3 and 6; why is it comparable in Table 3 but not in Table 6?
5. In PAL, how should one choose between LAL and PLITS? Whisper shows better performance with PLITS, does this mean that LAL fails for speech-domain audio encoders?

---

> ### Author Response · Authors · 2025-11-28
> **Author Response to Reviewer qmDt (1/3)**
>
> Dear Reviewer qmDt,
>
> Thank you very much for your thoughtful and detailed review of our submission. Below, we respond to each of your points in detail and highlight the corresponding paper updates.
>
> ----------------
>
> **W1 (A)** The advantage of LAL over existing Flamingo-style architectures is not fully convincing, as the paper lacks efficiency comparisons with Flamingo-style baselines. + Moreover, Figure 1 only presents comparisons against PLITS baselines, without including Flamingo-style counterparts.
>
>
> Thank you for pointing this out. In the revised manuscript, we now include a **Flamingo-style (FI)** baseline in our efficiency analysis. Figure 1 has been updated to report training throughput and inference memory for **PLITS, Flamingo, LAL, and PAL** under the same backbone and encoder configuration, so the efficiency gains of LAL and PAL can be compared directly against both PLITS and Flamingo.
>
> **Paper update.** Figure 1 has been revised to include Flamingo-style baselines alongside PLITS, LAL, and PAL.
>
> ----------------
>
>
> **W1 (B)** If my understanding is correct, the claimed improvement over Flamingo-style architectures lies in bypassing the audio FFNs to improve efficiency. However, since the main computational bottleneck in PLITS arises from audio self-attention \(O(N_a^2)\), it is unclear whether modifying the audio FFNs meaningfully contributes to reducing the overall time complexity by another \(O(N_a)\).
>
>
> Thank you for this question. Our gains over Flamingo-style architectures and PLITS come from both **information-flow changes** and **complexity reductions**, not just from skipping audio FFNs.
>
> **Comparison to Flamingo.** In Flamingo, text tokens first pass through an extra cross-attention + FFN block that attends to audio tokens; this output is **added back to the text residual stream** and then processed by the usual self-attention + FFN stack of the LLM. In LAL, encoder features are projected once per layer and text tokens attend **jointly** over text and audio within the *existing* LLM's self-attention block, with no additional cross-attention or FFN adapter layers. This defines a **different information pathway** and removes the entire Flamingo cross-attention stack, which yields a substantial efficiency gain.
>
> **Comparison to PLITS.** In PLITS, audio tokens are promoted to full LLM tokens and at every layer they undergo causal self-attention and FFNs, so both the compute and the information flow are the same as for text. LAL instead keeps a direct link to the encoder output: each layer has a small projector that maps “semantic-ready” audio features into that layer’s space, and these projected tokens are used only as **keys/values**. Audio tokens no longer query each other or go through FFNs.
>
> On the complexity side, PLITS uses full self-attention over all tokens with cost
> O((N_a + N_t)^2),
>
> while LAL lets only text tokens issue queries, reducing this to
> O((N_a + N_t) * N_t),
>
> which removes the O(N_a^2) audio–audio term. Since typically N_a >> N_t (for example,
> N_a ≈ 1500 and N_t ≈ 60), this is a large reduction: roughly from 1500^2 operations
> to about 1500 * 60 attention operations, on top of skipping audio FFNs. This get significant with long duration audio.
>
>
> **Paper update.** We have added a dedicated subsection *“Information Injection Dynamics. LAL is neither Lite PLITS nor Lite Flamingo”* clarifying these architectural and pathway differences, and we now explicitly present the PLITS vs LAL complexity comparison and Flamingo baseline efficiency(Figure 1) results in the revised manuscript.
>
> ----------------
>
>
> **W2: The efficiency of PAL is not reported.** Figure 1 only shows comparisons between LAL and PLITS; how PAL performs relative to PLITS baselines remains unclear.
>
>
> Thank you for pointing this out.  Figure 1 has been updated to report training throughput and inference memory for **PAL** under the same backbone and encoder configuration, so the efficiency gains of **PAL** can be compared directly others.
>
> **Paper update.** Figure 1 has been revised to include **PAL** alongside PLITS, Flamingo and LAL.

---

> ### Author Response · Authors · 2025-11-28
> **Author Response to Reviewer qmDt (2/3)**
>
> **W3 PLITS vs LAL for encoders** The criterion for selecting between LAL and PLITS in PAL is ambiguous. When the audio encoder differs from those used in the paper, it is unclear which variant (LAL or PLITS) would be more suitable or likely to perform better.
>
> We agree with this observation. Our encoder integration choices are primarily empirical, and we do not propose a criterion for deciding, given a particular encoder, whether LAL or PLITS is more appropriate.
>
> At the same time, we hope the reviewer agrees that, in the current landscape, exploring how to integrate multiple modalities with LLMs through hybrid integration is already valuable in its own right as more and more modaliteids are bing integrated to LLM for omin setups. To the best of our knowledge, our work is the first to propose such a hybrid LLM based integration within this setting, and we view a more systematic treatment of encoder specific integration strategies as an important direction for future work.
>
>
> ----------------
>
>
>
> **W4 The writing could be improved.** Section 3 (Methodology) mixes method and results, which should be clearly separated. In addition, the structure of PAL in Figure 2(C) is not fully explained in the text and lacks proper legends.
>
>
> We thank the reviewer for this helpful feedback regarding the writing and organization. Our original intention in Section 3.3 was to present the architectural evolution of LAL and PAL together with the empirical evidence that motivated the final PAL design, but we agree that this mixed presentation made the section harder to follow.
>
> **Paper update.**
> * We have substantially revised the structure of the paper to clearly separate **Methodology** and **Experiments and Results**. All descriptions of LAL, PAL and their variants are now contained within the Methodology section, and the corresponding empirical evaluations are moved to the Experiments and Results section. This avoids mixing method and results and improves readability and reproducibility.
>
> * In addition, we have expanded the textual explanation of the PAL structure corresponding to Figure 2(C), and we have revised the figure legends to more clearly explain each component and connection.
>
> We believe these changes directly address the reviewer’s concerns and improve the clarity of the presentation.
>
>
> ----------------
>
>
>
> **Q1** How efficient are LAL and PAL compared with Flamingo-style architectures?
>
> Thank you for this question. To address it, we implemented a Flamingo style integration in our framework and updated Figure 1 to report its training and inference throughput and memory usage alongside PLITS, LAL, and PAL. Our analysis shows that LAL is more efficient than Flamingo, while PAL is slightly less efficient than Flamingo. From a speech understanding perspective, however, we do not expect Flamingo style integration to outperform LAL on speech understanding, and PLITS remains the stronger choice for this purpose. As suggestive external evidence, the state of the art Audio Flamingo series exhibits a similar pattern: Audio Flamingo 1 and 2 use a Flamingo style architecture, whereas Audio Flamingo 3, which places more emphasis on speech understanding, adopts a PLITS style integration completly discarding the flamingo integration. The paper does not explicitly attribute this change to efficiency or speech understanding, so we present this only as an interpretation and apologize if our reading of their design choice is inaccurate.
>
>
> **Paper update.** Figure 1 has been revised to include **Flamingo** alongside PLITS, LAL and PAL.
>
> ----------------
>
>
> **Q3** In Figure 2, what do the different input square colors (purple, pink, and blue) represent?
>
> We thank the reviewer for pointing this out. We have revised Figure 2 to improve the legend and clarify the meaning of the colored input squares. In the updated figure, each color group explicitly corresponds to a different type of token based on its integration type(PLITS, LAL, Flamingo), and this mapping is now clearly explained in the caption and legend.
>
> **Paper update.**
> Revised Figure 2 and its legend so that the roles of the purple, pink, and blue tokens are explicitly defined.

---

> ### Author Response · Authors · 2025-11-28
> **Author Response to Reviewer qmDt (3/3)**
>
> **Q4** In Tables 6 and 7, which models are considered fair comparisons? Some baselines outperform PAL but are not boldfaced. Does this mean they are not comparable, and if so, why? For example, Audio Flamingo-2-3B is listed in Tables 3 and 6; why is it comparable in Table 3 but not in Table 6?
>
> We thank the reviewer for this question and for catching the potential confusion.
>
> In Tables 6 and 7, **boldface is reserved only for models that are directly comparable under a strictly controlled setting**, where we keep the following fixed: the same LLM backbone family and size, the same audio encoders, the same training data, and the same tuning recipe. Concretely, the bold rows correspond to our own PLITS, LAL, and PAL variants that we train within this standardized pipeline. These are the models we regard as fair baselines for isolating the effect of the integration strategy.
>
> Although we maintain this controlled setting for our main comparisons, we agree that model size and training data quality and scale are the dominant factors when comparing to external systems. If one focuses on parameter count alone, Tables 6 and 7 already show that our 1B models outperform Audio Flamingo 2 3B and several 7B and 8B audio LLMs. This is exactly why we introduced our own PLITS baseline under the same encoders, backbone, and data as LAL and PAL so that the impact of the integration mechanism can be evaluated under matched conditions, rather than relying only on cross paper comparisons where data and training pipelines differ.
>
> Regarding the specific question about Table 3 versus Tables 6 and 7. Table 1 and Table 3 serve different purposes.
>
> * **Table 1** focuses on comparing LAL with PLITS and Flamingo style integration under controlled configurations within our framework.
> * **Table 3** is intended to situate LAL in the broader state of the art on OpenAQA style benchmarks. Here we compare LAL to external models such as LTU and GAMA that are also trained on OpenAQA, so there is at least some alignment in task and data, even though the overall training corpora and recipes are not identical.
>
> In contrast, **Tables 6 and 7** move to a different setting: they include speech understanding benchmarks and use a subset of AudioSkills (about six million samples) for training our models. Many of the external systems in these tables are trained on much larger and often proprietary datasets, so the data scale and domain mismatch is substantial. In this context, Tables 6 and 7 are primarily used to compare **PLITS versus PAL** under our standardized training recipe, with other models (including Audio Flamingo 2 3B) reported only as external references rather than controlled baselines. For that reason, those external rows are not boldfaced even if they sometimes achieve higher absolute scores.
>
>
> ----------------
>
>
> **Q5** In PAL, how should one choose between LAL and PLITS? Whisper shows better performance with PLITS, does this mean that LAL fails for speech-domain audio encoders?
>
>
> Our encoder integration choices are primarily empirical, and we do not propose a general rule for deciding, given a particular encoder, whether LAL or PLITS is more appropriate. In our current experiments on speech based emotion recognition and gender classification with Whisper, PLITS performs better than LAL, as shown in Table 11.
>
> During the rebuttal we also evaluated PLITS, LAL, and PAL on MMSU and added LAL results to MMAU and MMAR in Tables 6 and 7 (see speech focused subsets). These additional experiments show a similar pattern and further support the view that PLITS is the preferable choice for speech oriented encoders in our framework. However, we do not interpret this as LAL failing for speech. Rather, PLITS appears better as it is benefiting from the decoding inside the LLM. PAL is therefore designed to use PLITS for speech encoders such as Whisper while retaining LAL for general audio encoders.
>
> **Paper update**
> We have added a new discussion in Appendix E 3 that summarizes these observations on chosing PLITS vs LAL for speech.
>
> -------------------------------------------------
>
> **Q2 Is PAL a new architecture or .. LAL and PLITS  ..  encoder?**
>
> PAL is a hybrid architecture that uses PLITS for speech understanding, as supported by additional experiments in Appendix E3, and LAL for general audio understanding. In our multi encoder PAL configuration, Whisper features are integrated via PLITS, while the other audio encoders are integrated via LAL. In our unified PAL variant, we generate a set of summary tokens that are integrated via PLITS to preserve speech understanding, while the full sequence of audio tokens is integrated through LAL.
>
> We have updated Figure 2 to include token legends that indicates which tokens follow PLITS and which follow LAL.
>
>
>
> ------------------
>
> We sincerely appreciate your feedback and valuable suggestions, and we would be grateful if you could review the updated manuscript that now reflects the changes discussed above.

---

### Official Review · Reviewer_3Ffd · 2025-10-30

**Soundness:** 3
**Presentation:** 3
**Contribution:** 4
**Rating:** 8
**Confidence:** 3

**Summary:**

This paper addresses the problem of efficiently integrating audio encoders into LLMs for multi-modal understanding and reasoning. the authors introduce LAL (Lightweight Audio-LLM Integration), a new mechanism that injects audio representations only through the attention mechanism of LLMs (as keys and values), bypassing the feedforward submodules entirely. This design achieves significant efficiency gains, while maintaining or improving performance compared to PLITS. Experiments across multiple LLM backbones (Llama3.2-1B/3B, Qwen2.5-1.5B) and tasks (classification, captioning, and reasoning) demonstrate that PAL achieves comparable accuracy to PLITS but with substantially lower compute and memory cost.

**Strengths:**

- LAL introduces a clean and elegant modification to the transformer architecture (injecting audio only via attention keys/values) offering a new efficiency–performance trade-off frontier. This idea is both conceptually simple and practically powerful.
- The authors perform extensive and controlled experiments under identical training setups, ensuring fairness. They demonstrate consistent performance improvements or parity across classification, captioning, and reasoning tasks while greatly improving efficiency.
- The introduction of PAL provides a principled encoder-aware fusion that selects integration strategies based on encoder type, offering insight into modality-specific alignment (speech vs. general audio).
- The proposed architecture achieves major computational benefits (3.5× faster training, 83% less inference memory), making it especially appealing for future resource-efficient audio LLMs.

**Weaknesses:**

- Since PAL’s performance depends on multiple pretrained backbones, the generalization to new or smaller encoders remains to be tested.
- The paper relies primarily on embedding-based and GPT-4–based automatic evaluation; perceptual listening or human-judged reasoning studies could strengthen claims of interpretability and semantic alignment.
- While efficiency is well-quantified, the semantic fidelity of the transferred audio information (e.g., fine-grained temporal cues) under LAL is not deeply analyzed.

**Questions:**

- How does the optimal layer depth for audio injection vary between encoders (SSLAM, CLAP, Whisper)? Would adaptive layer selection improve the balance between efficiency and expressivity?
- Does LAL preserve temporal ordering and sound event localization cues as effectively as PLITS, given that audio-to-audio interactions are removed?
- The results on MMAR and MMAU show parity with PLITS but not consistent improvements. Are there cases where the hybrid PAL underperforms due to conflicting encoder fusion?
- Since LAL changes the forward architecture, have you considered combining it with parameter-efficient adapters or LoRA layers for even more lightweight fine-tuning?

---

> ### Author Response · Authors · 2025-11-28
> **Author Response to Reviewer 3Ffd  (1/3)**
>
> Dear Reviewer 3Ffd,
>
> Thank you very much for your thoughtful and positive review of our work. In the revised manuscript and rebuttal, we address your main concerns by (i) adding unified encoder PAL experiments with AF Whisper to test generalization beyond the original multi encoder setup, (ii) clarifying how LAL preserves temporal structure and semantic information from the audio encoders, and (iii) discussing extensions such as adaptive layer usage and combinations with LoRA for parameter efficient tuning. Below we respond to each of your weaknesses and questions in turn.
>
> --------------------------------------
>
>
> **W1: Since PAL’s performance depends on multiple pretrained backbones, the generalization to new or smaller encoders remains to be tested**.
>
>
> We thank the reviewer for raising this question about generalization beyond the specific multi-encoder configuration used in the initial submission.
>
> In the revised manuscript, we explicitly test PAL in a **unified-encoder** setting using a new audio backbone, AF-Whisper, to demonstrate that the approach is **not** tied to a fixed set of multiple backbones:
>
> * We introduce **PAL-UniEnc** in **Section 3.3.2**, where we use a *single* audio encoder (AF-Whisper) to handle both speech and general audio. PAL-UniEnc applies **PLITS only to a compact set of speech-summary tokens**, while integrating the full AF-Whisper token sequence via **LAL**, following the same hybrid principle as multi-encoder PAL.
> * We evaluate PAL-UniEnc-3B against a matched **PLITS-UniEnc-3B** baseline on **MMAU**, **MMAR**, and **MMSU** (Tables 5, 6, and 11). Across these benchmarks, PAL-UniEnc consistently matches or improves over the PLITS-UniEnc baseline while using a more efficient integration pattern. This shows that PAL’s benefits are not limited to a specific multi-encoder design and do transfer to a new unified encoder.
>
> We hope these additional experiments address the reviewer’s concern about generalization.
>
> **Paper update.** We added the unified-encoder PAL design and results in **Section 3.3.2**, Appendix and updated **Tables 5, 6, and 11** accordingly.
>
> --------------------------------------
>
>
> **W2**: The paper relies primarily on embedding-based and GPT-4–based automatic evaluation; **perceptual listening or human-judged reasoning studies could strengthen claims** of interpretability and semantic alignment.
>
> We appreciate this suggestion and agree that human perceptual studies would provide stronger evidence for interpretability and semantic alignment. Due to time and resource constraints during the rebuttal period, we could not run a user study.
>
> --------------------------------------
>
> **W3:** While efficiency is well-quantified, the semantic fidelity of the transferred audio information (e.g., fine-grained temporal cues) under LAL is not deeply analyzed.
>
>
> Thank you for raising this point. While we do not perform explicit representational probes, the standard audio event classification and captioning benchmarks in Table 1 already provide an indirect evaluation of semantic fidelity, since they require the model to preserve and use detailed semantic information from the audio.
>
> In addition, LAL is explicitly designed to preserve temporal ordering and sound event localization within the LLM. For a more detailed explanation of how temporal structure is maintained, please see our response to your **Question 2 (Q2)**.
>
>
> --------------------------------------

---

> ### Author Response · Authors · 2025-11-28
> **Author Response to Reviewer 3Ffd  (2/3)**
>
> **Q1:** How does the optimal layer depth for audio injection vary between encoders (SSLAM, CLAP, Whisper)? Would adaptive layer selection improve the balance between efficiency and expressivity?
>
> This is an interesting question. In our current work, we did not perform a systematic search over injection depth for each encoder (SSLAM, CLAP, Whisper), so we cannot claim an optimal layer depth per encoder.
>
> However, LAL is naturally suited to this type of study, since it allows us to choose which LLM layers receive audio features in a more flexible way than PLITS. Intuitively, one could route higher level semantic information such as events or intent to later LLM layers, while lower level cues such as pitch or prosody are injected into earlier layers.
>
> We agree that an adaptive, encoder dependent layer selection scheme could help trade off efficiency and expressivity, and we view this as a promising direction for future work rather than a question that is fully answered in the current submission.
>
>
> --------------------------------------
>
> **Q2:** Does LAL preserve temporal ordering and sound event localization cues as effectively as PLITS, given that audio-to-audio interactions are removed?
>
>
> We thank the reviewer for this question. Our design explicitly preserves temporal ordering in LAL, even though audio–audio attention inside the LLM is removed.
>
> * **RoPE position IDs.** In PLITS, the system prompt, audio tokens, and user text are concatenated and assigned contiguous position IDs (system prompt -> audio -> user input text). In LAL, we keep **exactly the same positional layout**: we reserve a block of position IDs for audio tokens and shift the user-text positions accordingly. These reserved audio positions are used when forming Keys/Values in the attention module, so text Queries still “see” audio tokens at the same relative RoPE positions as in PLITS. Thus, the temporal order encoded by the audio encoder is preserved in the LLM’s attention geometry. We have updated the manuscript to discuss this in detail with visualization (please see Appendix E4).
>
> * **LFST and temporal structure.** On the encoder side, LFST preserves the time axis when fusing SSLAM and CLAP by inserting a special “new-time-step” token before flattening the time–frequency grid. This ensures that each latent token sequence retains the original temporal ordering before it is passed to LAL; this mechanism is already described with visualization(Figure 3) in detail in **Appendix E.1**
>
> Now for the newly added, **Unified PAL (AF-Whisper).** the AF-Whisper tokens are kept at full temporal resolution and used directly as LAL Keys/Values, while PLITS summary tokens are interleaved in the K/V sequence in a way that keeps each summary token aligned with its underlying temporal span. This is discussed in detail with visualization(Figure 5) in Section  and Appendix E3.
>
> Together, these mechanisms ensure that temporal ordering and localization cues from the audio encoders are preserved for the text tokens attending under LAL; we only remove redundant **audio→audio** mixing inside the LLM, not the time structure itself.
>
> **Paper update.**  We have updated the manuscript with,
> * discussion on position id adjustments in **Appendix E.4**.
> * temporal preservation in Unified Encoder PAL is discussed in Section 3.3.2 and **Appendix E.3**
>
> --------------------------------------

---

> ### Author Response · Authors · 2025-11-28
> **Author Response to Reviewer 3Ffd (3/3)**
>
> **Q3:** The results on MMAR and MMAU show parity with PLITS but not consistent improvements. Are there cases where the hybrid PAL underperforms due to conflicting encoder fusion?
>
> Thank you for raising this point.  On MMAR and MMAU in the **multi encoder** setup, PAL is generally on par with PLITS and often slightly better on the overall and mixed modality aggregates, while delivering clear gains in efficiency. The small fluctuations across categories  **are not concentrated in any particular modality**.
>
> To further check that this behavior is not an artifact of multi encoder fusion, we added a **unified encoder PAL** variant using AF Whisper, where PLITS is applied only to summary tokens and LAL over the full sequence. In this single encoder setting we still observe small fluctuations in MMAR and MMAU, despite the absence of any potential encoder conflict, suggesting that these variations are not likely due to multi-encoder conflict effects.
>
> We also provide unified encoder experiments with AF-Whisper for both PLITS and PAL below for your convenience.
>
> **Table: Unified-encoder PAL vs. PLITS on MMAU-v05.15.25**
>
> | Model                      |  Sn mini  |  Sn test  |  Mu mini  |  Mu test  |  Sp mini  |  Sp test  | Total mini | Total test |
> | -------------------------- | ------- | ------- | ------- | ------- | ------- | ------- | :--------: | :--------: |
> | PLITS-UniEnc-3B (Baseline) |   75.68   |   72.03   | **70.96** |   69.63   |   46.25   |   46.48   |    64.30   |    62.91   |
> | PAL-UniEnc-3B (r=3) (Ours) | **76.28** | **73.87** |   69.76   | **70.03** | **49.25** | **54.46** |  **65.10** |  **66.26** |
>
> **Table: Unified-encoder PAL vs. PLITS on MMAR**
>
> | Model                      |     Sn    |     Mu    |     Sp    | Mix Sn–Mu | Mix Sn–Sp | Mix Mu–Sp | Mix Sn–Mu–Sp | Total Accuracy |
> | -------------------------- | ------- | ------- | ------- | ------- | ------- | ------- | :----------: | ------- |
> | PLITS-UniEnc-3B (Baseline) |   38.79   |   40.29   |   37.41   | **36.36** | **48.17** |   40.24   |   **50.00**  |      41.10     |
> | PAL-UniEnc-3B (r=3) (Ours) | **46.61** | **44.17** | **40.82** |   27.27   | **48.17** | **46.34** |     41.67    |    **44.40**   |
>
>
> --------------------------------------
>
>
> **Q4:** Since LAL changes the forward architecture, have you considered **combining it with parameter-efficient adapters or LoRA layers** for even more lightweight fine-tuning?
>
> Thank you for this suggestion. In the current work we do not combine LAL with LoRA or other PEFT modules. We follow a standard multimodal LLM pretraining and finetuning protocol: connector/projector pretraining in Stage 1, and joint finetuning of the connector/projector and the LLM in Stage 2. This setup keeps the comparison between PLITS, Flamingo, and LAL focused purely on the integration architecture.
>
> That said, LAL is designed to be orthogonal to PEFT methods. As we note in Section 3.2.1, techniques like LoRA change how weights are adapted during training while leaving the forward compute essentially unchanged, whereas LAL changes how audio tokens participate in attention and FFN routing, so its compute and memory savings also apply at inference.
>
> In principle, one could also apply LoRA on top of LAL for further fine tuning. Unlike a typical PLITS setup, where one might assign similar LoRA ranks to all query, key, value, and FFN weights, LAL naturally suggests a more targeted allocation: for example, using a **higher LoRA rank for the key, value, and LAL projector matrices that interact most directly with audio features**, and a smaller rank for the remaining weights. Exploring such LAL+LoRA configurations is an interesting direction for future work.
>
>
> --------------------------------------
>
> We sincerely appreciate your constructive suggestions and kindly invite you to review the updated manuscript, which now incorporates the changes discussed above.

---

### Official Review · Reviewer_HHJ6 · 2025-10-30

**Soundness:** 2
**Presentation:** 2
**Contribution:** 2
**Rating:** 4
**Confidence:** 3

**Summary:**

The paper proposes LAL and PAL, two efficient integration strategy to connect audio encoders with LLM. The core contribution is an architectural change called LAL, where audio tokens are only appended to keys and values in attention layers of LLM backbone. Through experiments, the authors propose to use LAL for audio encoders SSLAM and CLAP, and use traditional PLITS method for speech encoder Whisper. This hybrid strategy is called PAL. PAL significantly improves throughput and reduces GPU memory usage in both training and inference, while achieving comparable or better audio understanding performance than traditional PLITS method.

**Strengths:**

1. The proposed method LAL significantly boosts training throughput and reduces inference memory, while surpassing traditional PLITS method in many audio understanding and reasoning tasks.
2. The paper provides some valuable insights on how audio inputs interreacts with LLM backbone, explains how LLM digests features from different audio encoders, and proposed an encoder aware hybrid to balance efficiency and performance.

**Weaknesses:**

1. The idea of attending text tokens to audio tokens lacks novelty. The proposed LAL seems like a lightweight version of the "Flamingo style", where additional FFN removed and the cross and self-attention layer are combined into a single attention layer to improve efficiency.
2. The PAL seems more like a design choice based on empirical results, rather than an architectural change. The method seems more likely to be “hybrid” rather than “audio encoder aware”, since there lacks an adaptive algorithm to choose the probing method for each type of encoder. Therefore, it remains unclear if PAL can be extended to other audio encoders.
3. A typo for improved readability:
    - Line 170: the integration approach and the corresponding audio-LLM → the integration approach between audio encoders and the corresponding audio-LLM

**Questions:**

1. **Efficiency comparison for PAL**: Is there any comparison on throughput or GPU memory usage between PLITS and PAL rather than merely LAL?
2. **Usage of LFST**: Table 1 indicate that LFST is for achieving SOTA performance. Is LFST also applied in PAL experiments? Additionally, it would be better if results with PLITS and LFST could be provided when using Llama3.2-3B and Qwen2.5-1.5B, so that the advantage of LAL over PLITS could be evaluated under SOTA settings.
3. **Comparison with Flamingo-style**: Can LAL and PAL achieve audio understanding performance  comparable to Flamingo-style baselines? How much computational cost could be reduced? As Flamingo-style approach is closer to LAL than PLITS, it would be better if head-on comparison with Flamingo-style baselines can be included.

---

> ### Author Response · Authors · 2025-11-28
> **Author Response to Reviewer HHJ6 (1/3)**
>
> Dear Reviewer HHJ6,
>
> Thank you very much for your careful review and constructive feedback on our submission. We are grateful that you highlighted both the efficiency benefits of LAL and the insights on how audio inputs interact with the LLM backbone. In our rebuttal and revised manuscript, we have aimed to directly address your main concerns: (i) clarifying the relationship between LAL, PLITS, and Flamingo and emphasizing why LAL is not just a “lite Flamingo”, (ii) explaining the empirically motivated design of PAL and what we mean by “encoder aware” integration, (iii) providing clearer efficiency comparisons including PAL and Flamingo style baselines, (iv) detailing the usage of LFST and adding PLITS+LFST vs LAL+LFST results at the 3B scale, and (v) correcting the typo you pointed out.
>
> Below we respond to each of your points (W1–W2, Q1–Q2) in turn and summarize the corresponding changes made to the paper.
>
> ---------------------------------------
>
> **W1. LAL and Flamingo:**
>
> >The idea of attending text tokens to audio tokens lacks novelty. The proposed LAL seems like a lightweight version of the "Flamingo style", where additional FFN removed and the cross and self-attention layer are combined into a single attention layer to improve efficiency.
>
>
> Thank you for raising the connection to Flamingo style architectures. We agree that it is important to clarify how LAL relates to both PLITS and Flamingo, and we have updated the paper accordingly by adding the subsection *“Information Injection Dynamics. LAL is neither Lite PLITS nor Lite Flamingo”*.
>
> Conceptually, LAL is not obtained by simply removing the FFN or merging cross and self attention from Flamingo. In PLITS, audio tokens are promoted to full LLM tokens and are repeatedly transformed by causal self attention and FFN layers. Their representations gradually drift away from the encoder output as they mix with the language model state. In LAL, by contrast, the audio encoder produces semantic ready features that are projected at every layer through a dedicated MLP into the appropriate abstraction level for that layer. These projected features are used only as keys and values, which preserves a direct and controllable link to the audio encoder semantics rather than turning them into generic LLM tokens.
>
> Relative to Flamingo, LAL also routes information differently. Flamingo first runs a separate cross attention block where text queries attend to frozen visual or audio tokens, **adds this signal back to the text residual stream,** and then feeds it through the standard text to text self attention. **In LAL, text tokens attend jointly over text and projected audio tokens in a single self attention operation**, and no extra cross attention plus FFN adapter blocks are introduced. This gives a **distinct information pathway** and a different computational trade off, while still enabling **in context conditioning on encoder features**.
>
> To summarize, LAL is similar to PLITS in that it performs in context injection and allows text tokens to attend over both audio and text tokens, and it is similar to Flamingo in that it injects information that has not been fully decoded inside the LLM. However, it is architecturally distinct from both: LAL is neither a “lite PLITS” nor a “lite Flamingo,” but rather a new information pathway for integrating audio encoders with LLMs.
>
> **Paper Update**: We have added a new subsection titled *“Information Injection Dynamics. LAL is neither Lite PLITS nor Lite Flamingo”* to clearly explain these architectural differences and how LAL departs from both PLITS and Flamingo.
>
> ---------------------------------------
>
> **W2. The PAL seems more like a design choice based on empirical results:**
>
> >rather than an architectural change. .. be “hybrid”  than “audio encoder aware”.. PAL .. extended to other audio encoders.
>
>
> Thank you for this thoughtful comment. We agree that our choice to use **PLITS for speech encoders is empirically motivated**, rather than a fixed decision that we made first and only then tested. In our experiments we consistently observed that speech oriented tasks favored PLITS, while general audio encoders such as CLAP and SSLAM achieved a better efficiency and performance trade off with LAL. This pattern led us to instantiate PAL as a simple hybrid: speech encoders (trained with next token or ASR style objectives) are routed through PLITS, and general audio encoders (trained for semantic or event level descriptors) are routed through LAL.
>
> In this sense, “encoder aware” means that in a multi encoder system the integration pathway is chosen based on the function and training signal of each encoder family, rather than through an additional learnable routing module. We agree that designing a fully adaptive routing mechanism would be an interesting direction for future work, but we view this kind of hybrid integration as a promising step forward, especially as multiple modalities are connected to LLMs in future omni models.

---

> ### Author Response · Authors · 2025-11-28
> **Author Response to Reviewer HHJ6 (2/3)**
>
> **W3. typo for improved readability:**
>
> >Line 170: the integration approach and the corresponding audio-LLM → the integration approach between audio encoders and the corresponding audio-LLM
>
> Thank you for pointing this out. This sentence was intended to indicate that we use the names **LAL** and **PAL** both for the integration strategy and for the resulting audio LLM instantiated with that strategy.  In the revised version we have removed this sentence to avoid ambiguity.
>
> ------------------------------
>
>
> **Q1. Efficiency comparison for PAL:**
>
> >Is there any comparison on throughput or GPU memory usage between PLITS and PAL rather than merely LAL?
>
> We have added the throuput and GPU memroy comaprision for PLITS, FLAMINGO, LAL and PAL in the paper (Figure 1)
>
> **Paper Update**: The Figure 1 in manuscript has been revised to include comparison with PAL and FLAMINGO.
>
> ------------------------------
>
> **Q2. A) Usage of LFST:**
>
> >Table 1 indicate that LFST is for achieving SOTA performance. Is LFST also applied in PAL experiments?
>
> Yes. For all **multi-encoder** experiments, including PLITS, LAL and PAL, we use **LFST** as the connector to combine encoder outputs from SSLAM and CLAP without increasing the token count. The Whisper speech encoder is connected sepratly.**LFST** is used only to combine SSLAM and CLAP informatin as discussed in the Appendix E1.
>
> For the **unified-encoder** PAL experiments (we added during rebuttal) with AF-Whisper (Section 3.3.2), LFST is not needed, since there is a single audio encoder.
>
>
> **Q2. B) PLITS + LFST on Llama3.2–3B.**
>
> >Additionally, it would be better if results with PLITS and LFST could be provided when using Llama3.2-3B and Qwen2.5-1.5B, so that the advantage of LAL over PLITS could be evaluated under SOTA settings.
>
> Following the reviewer’s suggestion, we additionally trained a **Llama3.2–3B** model with **PLITS + LFST** and report it in the revised Table 1, alongside the corresponding **LAL + LFST** configuration. For convenience, we reproduce the 3B block here:
>
>
>
> | LLM Backbone | PLITS | FI  | LAL | LFST | ESC50 | DCASE |  VS   |  FSD  | AS2M |  AC† |  CL† |  AC‡ |  CL‡ |
> |-------------|-----|:---:|:---:|:----:|-----|-----|-----|-----|:----:|:----:|:----:|:----:|:----:|
> | Llama3.2–3B | ✓     | ✗   | ✗   | ✓    | 84.60 | 46.16 | 59.15 | 43.29 | 15.00 | 0.70 | 0.38 | 17.90 | 12.03 |
> | Llama3.2–3B | ✗     | ✗   | ✓   | ✓    | **89.25** | **47.21** | **60.46** | **43.86** | **15.03** | **0.73** | **0.40** | **18.61** | **12.46** |
>
>
>
> These results further support our main claim: **under SOTA-style settings with LFST and a larger LLM, LAL consistently outperforms PLITS** across classification and captioning benchmarks, while retaining its efficiency advantages.
>
>
> Due to time and compute constraints during the rebuttal period, we were not able to train **Qwen2.5–1.5B** with the full PLITS+LFST and LAL+LFST configurations. We plan to run these additional experiments if time and resources permit.

---

> ### Author Response · Authors · 2025-11-28
> **Author Response to Reviewer HHJ6 (3/3)**
>
> **Q3.Comparison with Flamingo-style:**
>
> >Can LAL and PAL achieve audio understanding performance comparable to Flamingo-style baselines? How much computational cost could be reduced? As Flamingo-style approach is closer to LAL than PLITS, it would be better if head-on comparison with Flamingo-style baselines can be included.
>
>
> We thank the reviewer for requesting a head-on comparison with Flamingo-style integration. In the revised manuscript, we have implemented a **Flamingo-style (FI)** baseline within our own training pipeline and evaluated it under the same Llama3.2–1B backbone, audio encoders, data, and optimization settings as PLITS and LAL. As shown in Table 1, **LAL achieves audio understanding performance that is comparable to, and often better than, Flamingo**: for example, with LFST enabled, LAL outperforms Flamingo on ESC50, DCASE, FSD, and most captioning metrics, while Flamingo is slightly stronger on AS2M. Overall, LAL matches or exceeds Flamingo’s accuracy while being more efficient. In addition, **Figure 1** reports compute metrics: relative to Flamingo, LAL (and hence PAL’s LAL branch) achieves higher training throughput and lower GPU memory usage, both in training and inference (Flamingo lies between PLITS and LAL/PAL in speed and VRAM). PAL then combines this efficient LAL integration for general audio with PLITS for speech, and our MMAU/MMAR results show that it remains competitive with Flamingo-based systems despite using smaller backbones and less data.
>
> For convenience, the key Llama3.2–1B comparison from Table 1 is:
>
>
>
> | Integration | PLITS | FI  | LAL | LFST | ESC50 | DCASE |   VS   |  FSD  | AS2M |  AC† |  CL† |  AC‡ |  CL‡ |
> |------------|-----|---|---|----|-----|-----|------|-----|----|----|----|----|----|
> | Baseline   | ✓     | ✗   | ✗   | ✓    | 84.10 | 45.28 | 57.59  | 42.49 | 14.74 | 0.70 | 0.39 | 17.90 | 11.82 |
> | Flamingo   | ✗     | ✓   | ✗   | ✓    | 84.95 | 43.95 | 55.44  | 41.27 | **15.00** | 0.69 | 0.39 | 17.09 | 11.91 |
> | LAL (Ours) | ✗     | ✗   | ✓   | ✓    | **87.40** | **46.23** | **56.03** | **43.91** | 14.74 | **0.72** | **0.42** | **18.08** | **12.58** |
>
>
> These results support our claim that LAL (and consequently PAL) can deliver Flamingo-level or better audio understanding performance while reducing computational cost.
>
> We also believe that, specifically for **speech understanding**, PLITS is better suited than Flamingo-style integration. As suggestive evidence, the state-of-the-art Audio-Flamingo series follows a similar pattern: **Audio Flamingo 1 and 2** use a Flamingo-style architecture, whereas **Audio Flamingo 3**, which places more emphasis on speech understanding, switches to a PLITS-style integration. The paper does not explicitly explain this change, but it is consistent with our observation; we present this as an interpretation and apologize if our reading of their design choice is inaccurate.
>
> **Paper update.** We have added Flamingo-style experiments to **Table 1**, included Flamingo in the compute and memory comparison in **Figure 1**, added an architectural comparison of PLITS, Flamingo, and LAL in **Figure 2**, and expanded the discussion of their differences in Section 3.2 (*Information Injection Dynamics. LAL is neither Lite PLITS nor Lite Flamingo*).
>
> ------------------
>
> We sincerely appreciate your feedback and suggestions, and we kindly request you to review the updated manuscript, which now incorporates the changes discussed above.

---

### Official Review · Reviewer_wECV · 2025-10-31

**Soundness:** 3
**Presentation:** 3
**Contribution:** 2
**Rating:** 6
**Confidence:** 4

**Summary:**

This paper introduces PAL (Probing Audio Language Models), a framework that integrates audio into Large Language Models (LLMs) efficiently. The authors aim to improve the transfer of rich audio semantics from encoders to LLMs **without incurring heavy computational costs**. PAL utilizes Lightweight Audio LLM Integration (LAL) and a hybrid integration method (PAL) that intelligently chooses the integration strategy depending on the encoder used. LAL enhances efficiency by only involving audio tokens in the attention mechanism, skipping feed-forward modules. PAL combines both LAL and PLITS (Prepend to the LLM’s input token space), depending on the audio encoder, to balance performance and computational efficiency. Experimental results show that PAL achieves better or comparable performance to state-of-the-art models, while significantly reducing **memory usage** and **computation time**.

**Strengths:**

1. The authors claim that PAL reduces the computational cost of integrating audio encoders into LLMs while maintaining performance compared a fully PLITS integration-based system.
- Experimental results: PAL outperforms PLITS on classification, captioning, and reasoning tasks across several LLM backbones, achieving higher throughput and lower memory usage.
- Ablation results: In Fig 1, comparing LAL with PLITS shows that LAL reduces training memory usage by up to 64.1% and increasing training speed by up to 3.5x Faster.
- **Theoretical analysis**: PAL’s design is grounded in efficient attention routing, showing how it reduces complexity and computation costs without sacrificing performance.

2. The PAL framework is evaluated on several LLMs such as Llama3.2-1B, Llama3.2-3B and Qwen2.5-1.5B. It compares LAL with PLITS in classification, captioning, and reasoning tasks on Table 1 and Table 2. It proves the generalization and scalability of PAL.

3. The paper formally proves that LAL achieves superior computational efficiency and lower memory usage compared to PLITS. To balance efficiency and performance, this paper proposes PAL that defines how audio tokens interact with text in LLM layers.

**Weaknesses:**

1. The authors need to further address how LAL impacts the ASR task for speech.

2. PLITS should be considered a ceiling for LAL because when all audio tokens are input together with text into the LLM, they interact through attention. However, when audio tokens are used as KV (Keys and Values) and text as Q (Query) for Cross-Attention (CA), there may be a loss of information. However, in most of the authors’ experiments, LAL outperforms PLITS. Please explain this phenomenon.

3. Comparative experiments for PLITS, LAL, and PAL in the ASR task.

**Questions:**

See weakness part.

---

> ### Author Response · Authors · 2025-11-28
> **Author Response to Reviewer wECV (1/2)**
>
> Dear Reviewer wECV,
>
> Thank you very much for your careful and constructive review, and for finding the paper marginally above the acceptance threshold. We appreciate your positive assessment of our efficiency analysis, experimental breadth, and the PAL design, and we are grateful for the specific weaknesses you highlighted. Below we respond to each of your points in turn and describe the changes made in the revised manuscript.
>
>
> --------------------------------------
>
> **W1. The authors need to further address how LAL impacts the ASR task for speech.**
> & **W3. Comparative experiments for PLITS, LAL, and PAL in the ASR task.**
>
> Our current training mixture is not designed as a conventional ASR system. Instead, it focuses on **speech understanding rather than word-for-word transcription**. In practice, the model is trained to capture what is being said and in what context (e.g., “people are talking about going to Disney World in a minivan after picking up their friends and family from the railway station, and they are excited about it”), rather than to output an exact transcript. This limits the direct applicability of standard ASR metrics such as WER under our current setup.
>
> To directly address the reviewer’s request for speech transcription–level evaluation of PLITS, LAL, and PAL, we adopt MMSU as our primary benchmark. MMSU is a comprehensive spoken language understanding benchmark that includes content grounding tasks **where the model must select the correct transcription from several closely related alternatives**. These tasks explicitly probe transcript-level understanding, not just coarse semantic gist.
>
> Using MMSU, we compare PLITS, LAL, and PAL on both paralinguistic and linguistic dimensions. The table below summarizes performance across perception and reasoning categories:
>
>
> | Integration Style   | Para(Perception) | Lingu(Percep) | Lingu(Reason) | Para(Reason) | Perception | Reasoning | Overall |
> |-------|------------------|---------------|---------------|--------------|------------|-----------|---------|
> | PLITS | **33.86**           | **33.69**        | 58.47        | 45.67       | **33.76**     | 56.69    | **44.86**  |
> | LAL   | 33.56           | 30.32        | 51.08        | **46.27**       | 31.59     | 50.41    | 40.70   |
> | PAL   | 32.67           | 29.62        | **58.66**        | 45.97       | 30.81     | **56.90**     | 43.44  |
>
>
> This shows LAL alone is not optimal for ASR-style speech tasks, PAL is explicitly designed to handle this by **routing Whisper-style speech tokens through PLITS while keeping general audio on the LAL path**. Our MMSU results empirically validate this design choice.
>
> **Paper Update:** To make this clearer in the manuscript and to directly respond to W1 and W3, we have:
> Added a dedicated subsection on speech understanding capabilities of PLITS, LAL, and PAL in Appendix E.3.
> Included detailed MMSU results for all three integration styles.
> Complemented MMSU with additional speech-based emotion recognition and gender classification experiments, further probing speech-related performance.
>
>
> We hope this clarifies how LAL and PAL behave on ASR-like speech understanding tasks and why our final architecture treats speech differently from general audio.

---

> ### Author Response · Authors · 2025-11-28
> **Author Response to Reviewer wECV (2/2)**
>
> **W2. PLITS should be considered a ceiling for LAL**
>
> >because when all audio tokens are input together with text into the LLM, they interact through attention. However, when audio tokens are used as KV (Keys and Values) and text as Q (Query) for Cross-Attention (CA), there may be a loss of information. However, in most of the authors’ experiments, LAL outperforms PLITS. Please explain this phenomenon.
>
>
> We thank the reviewer for this insightful observation. While PLITS appears, in principle, to provide a “maximal” interaction channel by letting audio tokens participate in both attention and FFN, our view is that **PLITS and LAL are two qualitatively different ways of extracting information from the audio encoder**, rather than one being a strict relaxation of the other. As we discuss in the new *Information Injection Dynamics* paragraph in Section 3.2, they make different trade-offs in how they use the encoder’s representations.
>
> **Why LAL can outperform PLITS in practice**
>
> * **Avoiding feature drift.** In PLITS, audio tokens are repeatedly transformed by text-optimized FFNs and causal self-attention at every layer. This can cause the original encoder features (which are already semantically rich) to “drift” as they mix with the LLM’s internal states, potentially diluting event-level cues that matter for general audio tasks.
>
> * **Maintaining a direct semantic link.** In LAL, each layer uses a small projector to map “semantic-ready” audio features directly into the appropriate abstraction for that layer. This preserves a tight link to the encoder’s semantic space while still allowing text tokens to attend to audio at every layer. For tasks that hinge on explicit acoustic events (e.g., *“Which animal sound is heard?”*), keeping a stable, high-fidelity representation can be more beneficial than deeply transforming the audio tokens through the FFN stack.
>
> * **Different objective, not a weaker version.** PLITS is well suited for speech-like encoders whose tokens already behave like linguistic units, which we indeed observe for Whisper. LAL, in contrast, is better aligned with general-audio encoders trained to produce compact semantic descriptors rather than full token streams. In this regime, “asking” text tokens to query a stable audio context (LAL) can extract information more faithfully than forcing audio tokens to participate as full LLM tokens (PLITS).
>
> Because of these differences, we do **not** regard PLITS as a theoretical ceiling for LAL. Instead, the two schemes target different integration regimes: PLITS excels for language-like speech tokens, while LAL is better matched to general audio encoders and can therefore outperform PLITS on many of the benchmarks we consider.
>
> **Paper update.** We clarify this point in the revised manuscript by expanding the *Information Injection Dynamics* paragraph in Section 3.2 to explicitly contrast the information paths of PLITS and LAL and to state that PLITS is not a strict upper bound for LAL, especially in the general audio encoder regime.
>
>
>
> We sincerely appreciate your positive feedback and valuable suggestions. We kindly ask you to review the updated manuscript, which includes the changes discussed above.

---

### Official Review · Reviewer_nc72 · 2025-11-01

**Soundness:** 3
**Presentation:** 1
**Contribution:** 2
**Rating:** 2
**Confidence:** 4

**Summary:**

This paper introduces PAL, a framework for efficiently integrating audio encoders with large language models (LLMs) for audio-language tasks. The authors identify two dominant integration paradigms—PLITS (prepending audio tokens to text tokens) and Flamingo-style cross-attention—and propose a novel lightweight alternative called LAL (Lightweight Audio LLM Integration). LAL injects audio representations only through the attention mechanism (as keys and values) and skips the feedforward network (FFN) for audio tokens, significantly reducing computational and memory costs. The authors further propose PAL, a hybrid system that uses LAL for general audio encoders (e.g., SSLAM, CLAP) and PLITS for speech encoders (e.g., Whisper), based on the observation that speech benefits from deeper integration. Extensive experiments across multiple LLMs and audio tasks show that LAL and PAL achieve competitive or superior performance to PLITS while offering lower memory usage and higher throughput.

**Strengths:**

1. LAL is a simple yet effective architectural modification that reduces both attention complexity and FFN computation for audio tokens, which is novel in the LALM training.
2. The authors provide a principled rationale for using PLITS with Whisper (speech) and LAL with general audio encoders, supported by empirical results and a neuro-linguistic analogy.
3. The paper conducts controlled comparisons under a standardized training curriculum, ensuring that performance differences are attributable to the integration method rather than data or model size. Evaluations span multiple LLMs (Llama 3.2 1B/3B, Qwen2.5 1.5B) and diverse audio tasks (classification, captioning, reasoning).

**Weaknesses:**

1. Unclear Presentation and Methodological Exposition: The paper suffers from unclear writing, particularly in Section 3.3 (Methodology), where the description of the proposed method is entangled with experimental details and results. This convoluted structure makes it difficult for the reader to cleanly grasp the core architectural innovations of LAL and PAL before being presented with empirical outcomes, hindering the understanding and reproducibility of the proposed approach.
2. Incomplete and Unconvincing Experimental Validation: The experimental validation is insufficient to robustly demonstrate the scalability and competitiveness of the proposed methods. While early results (e.g., Table 1 and Table 3) include a 3B parameter model, the final, crucial comparisons with state-of-the-art models (Tables 6, 7) are only performed with the 1B version. This raises a significant question: does the performance advantage hold when models are scaled? A fair and convincing comparison requires evaluating the 3B LAL/PAL models against other open-source audio-LLMs of similar scale (many of which have 3B variants).
3. Questionable Claims on Overall System Efficiency: The efficiency claims are narrowly focused on comparing LAL against the PLITS integration paradigm. However, the proposed PAL system employs two encoders (e.g., SSLAM/CLAP and Whisper), which inherently leads to longer input sequences and higher computational load from the encoder side. The paper does not demonstrate that PAL is more efficient than a streamlined, single-encoder model (such as Qwen-Audio or Qwen-Omni) that uses a more efficient integration method. Therefore, the claim of superior efficiency is valid only within the specific context of multi-encoder systems and may not hold against more optimized, holistic architectures.

**Questions:**

Refer to the weaknesses for the main concerns. The unclear presentation, incomplete and unconvincing experimental validation, and the questionable claims of efficiency lead me to reject the paper.

---

> ### Author Response · Authors · 2025-11-28
> **Author Response to Reviewer nc72 (1/2)**
>
> Dear Reviewer nc72,
>
> Thank you for your careful reading of our paper and for the thoughtful, detailed feedback. We have updated the PDF manuscript to address your main concerns regarding clarity of the methodology, scalability and competitiveness of LAL/PAL, and the scope of our efficiency claims. Below, we respond to each point (W1–W3) and highlight the corresponding changes in the revised paper.
>
>
> ----------------------------
> **W1. Unclear Presentation and Methodological Exposition:**
>
> >The paper suffers from unclear writing, particularly in Section 3.3 (Methodology), where the description of the proposed method is entangled with experimental details and results. This convoluted structure makes it difficult for the reader to cleanly grasp the core architectural innovations of LAL and PAL before being presented with empirical outcomes, hindering the understanding and reproducibility of the proposed approach.
>
>
> We thank the reviewer for this valuable feedback. Our original intention in Section 3.3 was to present the architectural evolution of LAL and PAL alongside the empirical evidence that supports the PAL design choice. However, we agree that intertwining methodology with experimental results made the core contributions harder to follow and could hinder reproducibility.
>
> **Paper Update:** We have substantially restructured the paper in response. The content is now organized into clearly separated **Methodology** and **Experiments and Results** sections. The descriptions of the LAL and PAL architectures are now fully self-contained within the Methodology section, ensuring that all model details are clear and reproducible before any results are introduced. We believe this revised structure significantly improves clarity and readability, and we invite the reviewer to examine the updated manuscript.
>
> ----------------------------
>
> **W2. A) Incomplete and Unconvincing Experimental Validation:**
>
> >The experimental validation is insufficient to robustly demonstrate the scalability and competitiveness of the proposed methods. While early results (e.g., Table 1 and Table 3) include a 3B parameter model, the final, crucial comparisons with state-of-the-art models (Tables 6, 7) are only performed with the 1B version. This raises a significant question: does the performance advantage hold when models are scaled?
>
> Thank you very much for raising this concern about scalability and the strength of our empirical validation. Our overarching goal in this work is to explore audio LLMs under multiple architectural integration strategies, and we designed our experiments with this in mind. In particular, Tables 1 and 3 report results at both 1B and 3B parameter scales, and we also evaluate LAL and PAL with two different families of base models, Llama 3.2 and Qwen 3.2, to support the claim that the proposed approaches generalize across architectures.
>
> Due to strict compute and time constraints during the rebuttal period, we were unfortunately not able to fully train and evaluate a 3B model on the complete set of benchmarks reported in Tables 6 and 7. However, the **newly added unified audio encoder experiments are conducted at the 3B scale, where both the PLITS baseline and PAL use Llama 3.2 3B**. These results indicate that the performance gains of PAL are not limited to the 1B configuration and do extend to 3B models as well (please see our response to W3).
>
> **W2 B) Comparision against other open-source audio-LLMs of similar scale:**
>
> >A fair and convincing comparison requires evaluating the 3B LAL/PAL models against other open-source audio-LLMs of similar scale (many of which have 3B variants).
>
> We already compare against an open source 3B model: **AudioFlamingo-2-3B** is included in Tables 6 and 7, and our 1B model **already outperforms this 3B system** on these benchmarks. More broadly, we believe the reviewer will agree that, for instruction tuned audio LLMs, performance depends not only on parameter count but also heavily on the **quality and scale of the training data** and the tuning recipe. In our experiments, the **our 1B models surpass several larger public audio LLMs (e.g., DeSTA2.5-Audio-8B, LTU-7B, GAMA-7B, SALMONN-7B, and AudioFlamingo-2)** despite their smaller size. This is precisely why we also train our **own PLITS baseline** under the same encoders, backbones, and data as LAL and PAL, so that the impact of the integration strategy can be assessed under fully controlled conditions.
>
>
> ----------------------------

---

> ### Author Response · Authors · 2025-11-28
> **Author Response to Reviewer nc72 (2/2)**
>
> **W3: Questionable Claims on Overall System Efficiency:**
>
> >The efficiency .. multi-encoder systems and may not hold against more optimized, holistic architectures.
>
>
> We agree that efficiency must be interpreted at the system level. Our claims are scoped: PAL is an *integration strategy*. We do not claim that PAL based systems are globally more efficient than highly optimized single encoder architectures such as Qwen Audio or Qwen Omni. All efficiency comparisons in the paper are made under a *fixed encoder–LLM configuration*.
>
> To address the concern about multi encoder overhead, we implemented PAL in a **single unified encoder** setting using AF-Whisper. PAL-UniEnc-3B applies PLITS only to a compact set of summary tokens (via a 1D Conv with stride (r)) and uses LAL over the full AF-Whisper sequence. This shows PAL as a pure integration mechanism in a streamlined, single encoder design and allows a direct comparison against PLITS-UniEnc-3B.
>
> Below we report throughput, memory, and average performance (MMAR, MMAU, MMSU):
>
> | Model               | Samples/s ↑ (Train / Inf) | VRAM (GB) ↓ (Train / Inf) | MMAR ↑    | MMAU ↑    | MMSU ↑    |
> | ------------------- | ------------------------- | ------------------------- | --------- | --------- | --------- |
> | PLITS-UniEnc-3B     | 70.68 / 7.80              | 42.49 / 17.68             | 41.10     | 62.91     | 40.12     |
> | PAL-UniEnc-3B (r=3) | 96.12 / 8.40              | 41.48 / 12.99             | **44.40** | **66.26** | **43.44** |
> | PAL-UniEnc-3B (r=5) | **105.72** / **8.76**     | **39.98** / **11.71**     | 42.00     | 63.42    | 41.22     |
>
> PAL UniEnc 3B with **r = 3** improves both efficiency and accuracy compared to PLITS UniEnc 3B. With more aggressive compression at **r = 5**, PAL UniEnc 3B becomes even more efficient while still outperforming the PLITS UniEnc 3B baseline on MMAR and MMSU and remaining competitive on MMAU. Note that the PLITS UniEnc 3B baseline processes **750** AF Whisper tokens. In contrast, PAL UniEnc 3B with **r = 3** processes 750 tokens through LAL plus (750 / 3) summary tokens through PLITS (1000 tokens in total), while with **r = 5** it processes 750 + (750 / 5) = 900 tokens. The stronger performance of the **r = 3** configuration is therefore consistent with its larger effective token budget.
>
>
> We also provide detailed MMAU, MMAR, and MMSU comparisons:
>
> **Table: Unified-encoder PAL vs. PLITS on MMAU-v05.15.25**
>
> | Model                      |  Sn mini  |  Sn test  |  Mu mini  |  Mu test  |  Sp mini  |  Sp test  | Total mini | Total test |
> | -------------------------- | ------- | ------- | ------- | ------- | ------- | ------- | :--------: | :--------: |
> | PLITS-UniEnc-3B (Baseline) |   75.68   |   72.03   | **70.96** |   69.63   |   46.25   |   46.48   |    64.30   |    62.91   |
> | PAL-UniEnc-3B (r=3) (Ours) | **76.28** | **73.87** |   69.76   | **70.03** | **49.25** | **54.46** |  **65.10** |  **66.26** |
>
> **Table: Unified-encoder PAL vs. PLITS on MMAR**
>
> | Model                      |     Sn    |     Mu    |     Sp    | Mix Sn–Mu | Mix Sn–Sp | Mix Mu–Sp | Mix Sn–Mu–Sp | Total Accuracy |
> | -------------------------- | ------- | ------- | ------- | ------- | ------- | ------- | :----------: | ------- |
> | PLITS-UniEnc-3B (Baseline) |   38.79   |   40.29   |   37.41   | **36.36** | **48.17** |   40.24   |   **50.00**  |      41.10     |
> | PAL-UniEnc-3B (r=3) (Ours) | **46.61** | **44.17** | **40.82** |   27.27   | **48.17** | **46.34** |     41.67    |    **44.40**   |
>
> **Table: Unified-encoder PAL vs. PLITS on MMSU**
>
> | Model                      |   P-Per   |   L-Per   |   L-Res   |   P-Res   |    Per    |    Res    |  Overall  |
> | -------------------------- | ------- | ------- | ------- | ------- | ------- | ------- | ------- |
> | PLITS-UniEnc-3B (Baseline) |   34.75   |   28.79   |   50.79   |   42.99   |   31.12   |   49.71   |   40.12   |
> | PAL-UniEnc-3B (r=3) (Ours) | **37.92** | **30.70** | **54.87** | **47.16** | **33.53** | **53.80** | **43.34** |
>
>
> In the unified AF-Whisper setup, both speech and general audio are embedded into a single token sequence, so the summary tokens are expected to compress *both* types of information. Currently, we use a single Conv1d layer to generate these summary tokens. We expect that a stronger summarization module (for example a lightweight Mamba-style block) could support larger reduction factors (r) with little or no loss of performance. We highlight this as promising future work.
>
> **Paper Update**
>
> To clarify the scope of our efficiency claims and respond to **W3**, we have:
>
> * Added **Section 3.3.2 (Unified Audio Encoder PAL)**.
> * Included the unified AF-Whisper experiments and corresponding tables in **Section 4.2** and the **Appendix** (Tables 5, 6, and 11).
>
> Thank you again for your valuable feedback. We kindly ask you to review the updated manuscript with the revisions outlined above.

---

### Author Response · Authors · 2025-11-28
**General Response: Summary of Manuscript Revisions**

Dear Reviewers and Area Chairs,

Thank you for your detailed analysis and constructive suggestions on our paper.

In response to your feedback and to improve the clarity, scalability, and robustness of our work, we have updated the manuscript accordingly. The revised PDF has been uploaded, and we kindly ask you to review it. **Please also note that the efficiency statistics in Figure 1 and throughout the document have been updated** to correct a calculation error present in the previous manuscript. Below is a summary of the changes made for your convenience:


**Reviewer nc72:** To address your concerns regarding presentation and scalability, we have: (1) substantially restructured the paper to clearly separate **Methodology** (Section 3) from **Experiments and Results** (Section 4), and (2) introduced **Section 3.3.2** (Unified Audio Encoder PAL) and provided results at the 3B parameter scale (Tables 5, 6, and 10) to demonstrate that PAL's efficiency and performance gains hold in scalable, single-encoder setups.

**Reviewer wECV:** In response to your questions about speech understanding and integration dynamics, we have: (1) added **Appendix E.3**, which provides a detailed analysis of speech tasks (including the MMSU benchmark) to justify the hybrid design of PAL, and (2) expanded Section 3.2 to clarify why PLITS is not a theoretical ceiling for LAL regarding general audio tasks.

**Reviewer HHJ6:** Based on your feedback regarding baselines and architectural novelty, we have: (1) updated **Figure 1** and **Table 1** to include direct efficiency and performance comparisons against a **Flamingo-style** baseline, (2) added a new subsection in Section 3.2 ("Information Injection Dynamics") to explicitly differentiate LAL from Flamingo and PLITS, and (3) provided results for **Llama3.2-3B** with PLITS+LFST in Table 1.

**Reviewer 3Ffd:** To address your points on generalization and temporal structure, we have: (1) added **Appendix E.4** and **Figure 6** to visualize how LAL preserves temporal ordering via RoPE position IDs, and (2) validated generalization to new encoders via the **Unified Encoder (AF-Whisper)** experiments in Section 3.3.2.

**Reviewer qmDt:** In response to your suggestions on efficiency comparisons and clarity, we have: (1) revised **Figure 1** to include training throughput and inference memory for **Flamingo-style** architectures, and (2) updated **Figure 2** and its legends to explicitly define the roles of the colored tokens and their correspondence to integration types.

We believe these updates improve the clarity and completeness of our work, and we look forward to your further feedback. Below, we address each reviewer’s concerns in detail.

---

### Meta-Review · Area_Chair_ZQqv · 2026-01-06

**Summary:**

This paper investigates different strategies for integrating heterogeneous audio information into audio-centric large language models (audio LLMs). The authors propose a lightweight and efficient LAL approach, which is computationally more efficient than the standard PLITS strategy that directly interleaves audio and text tokens. Furthermore, by introducing a hybrid PAL approach that combines PLITS and LAL, the resulting model achieves comparable performance on multiple downstream tasks while using a substantially smaller number of parameters.

However, several substantive concerns raised by the reviewers remain unresolved:

1. Reviewer nc72-W2: Additional validation on larger-scale LLM backbones (e.g., 3B and 7B parameters) for the results reported in Tables 6 and 7 is not provided. Comparisons against existing large audio LLMs remain potentially inconclusive due to differences in backbone architectures, audio encoders, and training data or configurations.

2. Reviewers wECV-W1 and W3: While the authors report results on the MMSU speech understanding benchmark in the rebuttal, direct evaluation on automatic speech recognition (ASR) tasks is still missing. Although word error rate does not always correlate linearly with higher-level speech understanding, ASR performance remains an important reference point. In many speech understanding scenarios, accurate recognition of keywords and contextual terms is critical, which may not be fully captured by MMSU alone.

3. Reviewers 3Ffd-W2 and W4: Due to time constraints, the authors were unable to incorporate human perceptual evaluations or explore adaptations using LoRA-based fine-tuning, leaving these concerns insufficiently addressed.

4. Reviewer qmDt-W1: The relation of token lengths N_t and N_a​ is task-dependent, yet the current evaluation excludes tasks requiring fine-grained textual outputs, such as ASR or speech-to-text translation, which limits the generality of the conclusions.

While the authors have made substantial improvements to the clarity and presentation of the paper in response to reviewer feedback, these revisions primarily address exposition rather than the core technical concerns. As such, the revised work would be more appropriate for a future resubmission followed by a full reevaluation.

**Reviewer Concerns:**

W2 from Reviewer nc72; W1 and W3 from Reviewer wECV; W2 and W4 from Reviewer 3Ffd; and W1 from Reviewer qmDt remain unresolved, while the remaining concerns have been adequately addressed.

**Reviewer Scores:**

I expect that Reviewers nc72, wECV, and qmDt would likely maintain their original ratings, either due to the remaining unresolved concerns or because their initial evaluations were already high. In contrast, Reviewer 3Ffd may be inclined to increase their rating, as the outstanding issues are comparatively less critical.

---

### Decision · Program_Chairs · 2026-01-26

Reject